

# Evaluation of the Snow CCI Snow Covered Area Product within a Mountain Snow Water Equivalent Reanalysis

Haorui Sun[1], Yiwen Fang[3], Steven A. Margulis[1], Colleen Mortimer[2], Lawrence Mudryk[2], Chris Derksen[2]

[1]Department of Civil and Environmental Engineering, University of California, Los Angeles, Los Angeles, CA 90095, USA
[2]Climate Research Division, Environment and Climate Change Canada, Toronto, Canada
[3]Zhejiang University-University of Illinois at Urbana-Champaign Institute, Haining, China

*Correspondence to*: Steven A. Margulis (margulis@seas.ucla.edu)

**Abstract.** An accurate characterization of global snow water equivalent (SWE) is essential in the study of climate and water resources. The current global SWE dataset from the European Space Agency Snow Climate Change Initiative is derived from

the assimilation of passive microwave satellite data and in situ snow depth measurements. However, gaps exist in the current Snow CCI SWE dataset in complex terrain due to difficulties in characterizing mountain SWE via the passive microwave sensing approach and limitations of the in situ snow depth measurements. This study applies a Bayesian snow reanalysis approach with the existing Snow CCI snow cover fraction (SCF) dataset (1 km resolution) to develop a SWE dataset over four mountainous domains in Western North America for WYs 2001-2019. The reanalysis SWE estimates are evaluated through

comparisons with independent SWE datasets, and a parallel SWE reanalysis generated using snow extent retrieved from Landsat imagery (30 m resolution). Biases in Snow CCI reanalysis SWE were diagnosed by comparing Snow CCI snow cover with the Landsat reference. Both the number of SCF images and their characteristics (such as zenith angle) significantly affect the accuracy of SWE estimation. Overall, the Snow CCI SCF inputs produce reanalysis SWE of sufficient quality to fill the mountain SWE gap in the current Snow CCI SWE climate data record. A better characterization of the SCF uncertainty and a

bias correction could further improve the accuracy of the reanalysis SWE estimates.

## 1 Introduction

Seasonal snowpack in mountainous regions plays a vital role in the global energy and water cycle. The unique properties of snow, such as its high albedo and low thermal conductivity, make it a significant factor in the energy budget of the lower atmosphere and the Earth's surface. The seasonal snowpack conditions also influence the local weather and

circulations of monsoon (Rudisill et al., 2021). Furthermore, mountains act as natural water reservoirs, often referred to as "water towers", by storing water in the form of snowpack and glaciers at high altitude (Immerzeel et al., 2020). It is estimated that between 50% and 70% of the annual precipitation in the mountainous regions of the Western United States (WUS) falls as snow and is stored in the snowpack. During warmer seasons, this stored water is released as snowmelt, crucial for meeting downstream water demands and sustaining ecosystems. Many major rivers worldwide such as the Colorado (spanning the

United States and Mexico), Indus (flowing through the Himalaya), and Mackenzie (in Canada), heavily rely on mountain





snowmelt. Such water towers are vulnerable to climatic and socio-economic changes, which can cause negative impacts on the ~2 billion people (22% of global populations) living downstream (Immerzeel et al. 2020; Mankin et al., 2015). Monitoring and managing the water resources from mountain snowpacks requires accurate snow water equivalent (SWE) information. However, recent studies indicate that large discrepancies exist in the climatology of seasonal SWE magnitude and timing

across different global datasets (Fang et al., 2023; Liu et al., 2022; Wrzesien et al., 2019; Mudryk et al., 2024; Kim et al., 2021) with uncertainty particularly high in mountain areas.

The European Space Agency (ESA) Snow Climate Change Initiative (Snow CCI) has developed homogenized, high quality long-term snow cover fraction and SWE datasets to contribute to the understanding of snow in the climate system. In its initial phase, the Snow CCI project focused on generating consistent multi-sensor time series of daily fractional snow cover

from optical satellite data (Nagler et al., 2022) and SWE derived from assimilation of passive microwave (PM) satellite data and in situ snow depth. Snow CCI SWE adapted the GlobSnow (v3) algorithm which estimates SWE by combining PM brightness temperatures centered on 19 and 37 GHz from Scanning Multichannel Microwave Radiometer (SMMR), Special Sensor Microwave/Imager (SSM/I), and Special Sensor Microwave Imager/Sounder (SSMIS) with in-situ daily snow depth measurements via Bayesian non-linear iterative assimilation (Luojus et al., 2021).

The Snow CCI SWE product (and predecessor GlobSnow versions originally described in Takala et al., 2011) does not provide data over complex terrain because the coarse grid spacing (12.5 to 25 km) of the Snow CCI SWE product is incompatible with the scales of SWE variability in complex terrain. The Snow CCI SWE retrieval approach of combining satellite passive microwave measurements with surface snow depth measurements is not well-suited to the complex terrain and deep snow typical of mountain regions because (1) the passive microwave sensitivity to SWE saturates when SWE exceeds

150 mm  (Chang et al., 1982, 1987), and (2) the available snow depth observations are too sparse to meaningfully capture elevation and topographic variability in snow depth distribution (Pulliainen, 2006).

This study is motivated by the need to fill the mountain SWE gap in the Snow CCI SWE product. Specifically, we explore the use of a previously implemented Bayesian snow reanalysis framework (BSRF; (Margulis et al., 2016, 2019; Cortés et al., 2016; Liu et al., 2021; Fang et al., 2022). The framework combines prior snow estimates from an ensemble of land

surface model simulations with satellite derived fractional snow covered area to generate retrospective time series of snow extent and SWE (Margulis et al. 2019). A snow cover fraction product derived from optical satellite data spanning multiple decades already exists within the Snow CCI program, which provides a natural connection to a mountain SWE product via the BSRF.

The primary objective of this study is to evaluate the use of Snow CCI SCF products in the BSRF for estimating SWE

in mountainous terrain at Snow CCI SCF grid resolution of 0.01 degrees (~ 1km), thereby potentially filling a key gap in the existing Snow CCI SWE product. This assessment focuses on two watersheds in the WUS where the BSRF was previously implemented and two watersheds in Canada where the BSRF is implemented for the first time. The Snow CCI SCF products are global in coverage, so this approach can potentially be extended to all mountain regions in the future. Section 2 describes



the methods, data, and application domain, Section 3 provides results and discussion, and Section 4 provides the key
conclusions of the study.

## 2 Data, Application Domains, and Methods

### 2.1 Description of the Snow CCI SCF product

The Snow CCI SCF dataset is globally available at a spatial resolution of 0.01° (~ 1 km). The 1 km resolution of the Snow CCI fSCA product is relatively coarse compared to the resolution of available optical imagery, but is well suited to the potential application across all northern hemisphere mountain areas for gap-filling the existing Snow CCI SWE product. This study uses the MODIS-based Snow CCI Daily SCF product (version 2), available over the period 2000-2020 (http://cci.esa.int/data). This product is based on data from the MODIS sensor aboard the Terra satellite (MOD021KM and MOD03). While the MODIS sensor provides radiance data at spatial resolutions of 250, 500, and 1000 m, the Snow CCI SCF uses data from the 1 km Level 1B dataset, which aggregates all radiance data to the largest spatial scale. The processing chain of Snow CCI SCF product includes (1) pre-processing of satellite data, (2) cloud screening, (3) binary snow pre-classification based on the Normalized Difference Snow Index (NDSI), and (4) SCF retrieval using the adapted SCAmod algorithm (Metsämäki et al., 2012, 2015) described in more detail in Nagler et al. (2022).

The product contains two SCF datasets: viewable snow cover (SCFV), which is the snow cover fraction in open areas and on top of vegetation cover, and snow cover on the ground (SCFG), which is the snow cover fraction in open areas (the same as SCFV) and under the forest canopies. The SCFV dataset is used in this study since only viewable snow cover (i.e., through forest gaps) is assimilated in the current snow reanalysis framework as described in Sect. 2.3. The Snow CCI SCFV dataset will be referred to as Snow CCI fSCA hereafter. The Snow CCI SCF products have an accompanying uncertainty layer. We tested the feasibility of using this layer to provide spatially and temporally varying weights within the assimilation framework but found the uncertainty layer to be poorly suited for this implementation. Specifically, it does not consider the impact of viewing angle geometry which is an important influence on MODIS-derived fSCA (Sect. 2.3.2) so the uncertainty values were too low for our data assimilation purposes such that the fSCA images were weighed too heavily which degraded the performance compared to the prior.

### 2.2 Applications domains

The test domains include four mountainous watersheds in Western North America: the Tuolumne basin and Aspen/Castle-Maroon (i.e., Aspen) in the WUS and the Lajoie basin and Bow River basin in western Canada (Fig. 1). These four watersheds are selected because (1) they are representative of snow-dominated mountainous domains that are masked out in the current Snow CCI SWE product (Fig. 1), (2) remotely-sensed and in-situ SWE reference datasets are available for verification purposes, and (3) the Lajoie and Bow River basins are higher-latitude forested basins that have not been explored in previous applications of the BSRF.



More specifically, the upper Tuolumne Basin is a high-elevation watershed in the Sierra Nevada of California (CA). The elevation ranges from approximately 1241 m a.s.l to 3700 m a.s.l. The precipitation is dominated by snowfall during the winter. The snowmelt during the spring and early summer fills the Tuolumne River, which is dammed at Hetch Hetchy to provide water for San Francisco. The Aspen is a high-elevation domain situated in the Roaring Forking watershed in central Colorado (CO) on the western side of the Continental Divide. The elevation varies within the range 2550 m a.s.l to 4100 m

a.s.l. The snowmelt feeds the Castle and Maroon Creeks and contributes to almost all of the water supply to the city of Aspen.

      The La Joie Basin is a watershed in the Coast Mountains of British Columbia (BC). The elevation within the La Joie Basin varies from 800 m a.s.l to 2800 m a.s.l. The forested areas at low elevations cover 47% of the total watershed area (Darychuk et al., 2023). The snowmelt is a critical source of downstream freshwater to Downton Lake formed by the La Joie dam. The Bow River Basin in Southern Alberta (AB) is a well-studied basin located in the Canadian Rockies. The elevation

spans from 1250 m a.s.l to 3500 m a.s.l .The snowmelt from the Bow River Basin provides around 80% of the Bow River streamflow, which is vital for hydroelectric power, irrigation, and significant downstream populations (Wang et al., 2019). Despite the value of water resources and the frequency of damaging flood events (Pomeroy et al., 2015), there is no systematic airborne or satellite snow monitoring program for the mountain regions of western Canada.

      The goal herein is to develop and evaluate a spatiotemporally continuous Snow CCI-derived SWE reanalysis dataset

over these four mountain watersheds. The snow reanalysis framework is applied over tiles of 1º latitude by 1º longitude. The tiles that cover the application domains are outlined in the location map (Fig. 1) and are used for the reanalysis application. In this study, we conducted the Snow CCI reanalysis for WYs 2001-2019 across the study domains, corresponding to the period of available Snow CCI fSCA data.





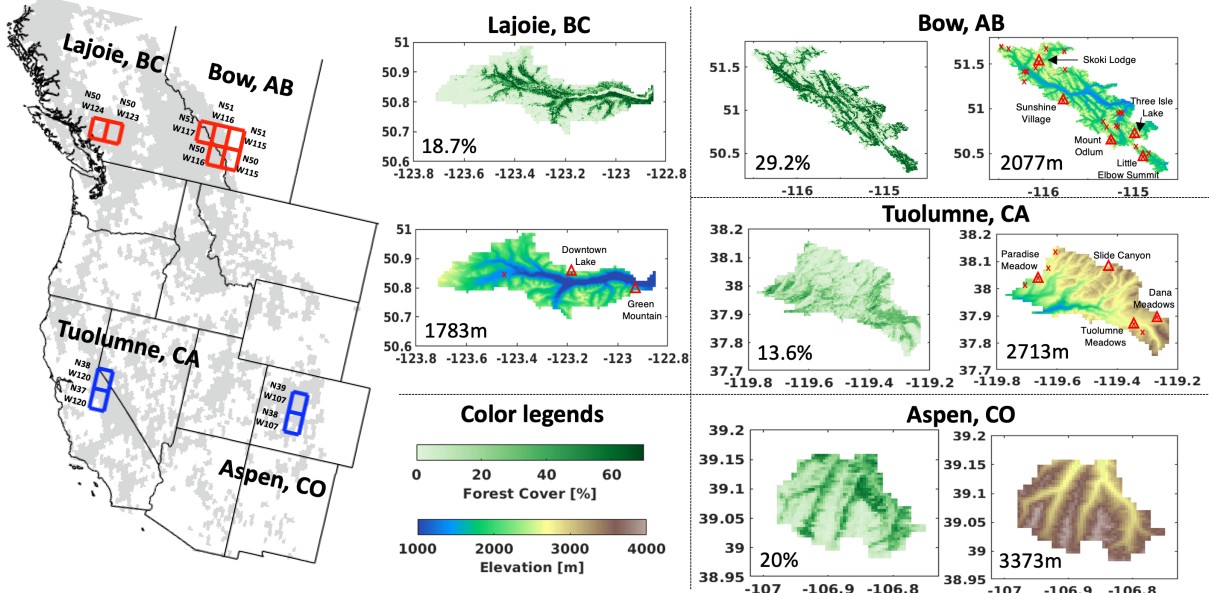


**Figure 1.** Location, forest cover, and elevation maps over the study domains, including Tuolumne basin (CA) and Aspen (CO) in the WUS., and Lajoie basin (BC) and Bow River basin (AB) in western Canada. The current Snow CCI SWE mask (that excludes mountainous areas) is shown in grey in the left panel. The test tiles (1° by 1°) that include the four study domains are outlined in red (Canada) and blue (WUS) and fall almost exclusively within the Snow CCI SWE mask. Degrees N and W

in left panel correspond to the lower left corner of each of these test tiles. The average elevation and forest cover are annotated in the spatial maps, where snow pillow locations are shown with red triangles and snow courses with red crosses.

## 2.3 Bayesian snow reanalysis framework

This study uses a previously developed Bayesian snow reanalysis framework (Margulis et al., 2019) to generate SWE

reanalysis estimates by assimilating the Snow CCI fractional snow-covered area (fSCA). The SWE reanalysis framework has been applied to generate snow reanalyses over the Sierra Nevada (Margulis et al., 2016), Andes (Cortés et al., 2016), High Mountain Asia (HMA) (Liu et al., 2021), and Western U.S. (Fang et al., 2022) using mostly Landsat-derived fSCA. These applications of SWE reanalysis have been verified against available in situ and airborne SWE estimates.

In this study, the land surface model inputs and uncertainty parameters for prior ensemble perturbations are following

Fang et al. (2022), which were derived specifically for the WUS. Historical remotely sensed fSCA measurements from Snow CCI are assimilated to update the prior estimates via a Particle Batch Smoother (PBS) scheme to yield SWE reanalysis estimates. The preprocessing of Snow CCI fSCA for SWE reanalysis including cloud screening (Sect. 2.3.1) and viewing





geometry screening (Sect. 2.3.2) processes are carefully conducted to avoid misclassifying cloud, forests, and other scatters as snow.

While previous applications of the SWE reanalysis via the assimilation of Landsat fSCA (with a native resolution of 30 m and scaled up ~100-500 m and a measurement error of 10%) have shown significant promise (Fang et al., 2022), herein we evaluate SWE estimates derived from the globally available Snow CCI fSCA product at 1 km resolution. We focused first on the western United States because previous well-validated implementations of the BSRF are available as the baseline. The implementation and validation over two watersheds in western Canada were performed in order to assess transferability to
new regions. The Landsat reanalysis is performed in parallel to provide a baseline for the comparison.

### 2.3.1 Cloud screening for Snow CCI fSCA data

The depletion of fSCA through the snowmelt season is informative to constrain the SWE evolution (Margulis et al., 2019) in both accumulation and melt seasons. Therefore, the PBS approach that assimilates time series of Snow CCI fSCA over the full WY (i.e., all images at once) is used to update the prior snow estimates at a single step. In previous SWE reanalysis
applications (over the WUS, Andes, and HMA), the internal cloud mask from the sensor and an ad-hoc cloud fraction threshold were used to identify images with "significant" cloud cover that were screened out entirely vs. those that were included, but with cloudy pixels within the image screened out. The tile-wise images with cloud fraction greater than the threshold were deemed "too cloudy" such that they might have significant misclassification of cloudy pixels as snowy pixels. Previously used cloud fraction thresholds were 0.4 for Landsat and 0.1 for MODIS-derived MODSCAG (Painter et al., 2009). There is a
tradeoff involved in using a single threshold. Opting for a higher threshold increases the pool of fSCA images, albeit with the risk of higher uncertainty such as mis-classifying clouds as snow pixels. On the other hand, selecting a lower cloud threshold results in screening out more images, potentially yielding fewer informative measurements but with an enhanced overall quality. Therefore, selecting the Snow-CCI cloud fraction threshold across test domains required careful consideration. Given that Snow CCI fSCA includes a unique internal cloud mask compared to other products, it is essential to derive a cloud fraction
threshold specific to the Snow CCI fSCA product.

The cloud fraction threshold was estimated through examination of the historical tile-wise cloud distributions (as diagnosed by the Snow CCI cloud mask) covering the test domains. Figure 2 shows the historical cumulative distribution functions (CDFs) (spanning WY 2000-2019) of tile-wise cloud fraction (aggregated by domains) for each WY in grey dashed curves and for the multi-year (WY 2000-2019) median in red solid curves. The median of the historical CDF (the intersection
of the blue dashed lines) is presumed to represent a typical tile-wise level of cloudiness in the binary classification of images into cloudy and less cloudy categories. However, there is a large variation in the median cloud fraction across different tiles/domains. It is apparent that the selected tiles in the WUS (Fig. 2a-b) are less cloudy than those in Western Canada (Fig. 2c-d). The CDF of the tile-wise median cloud fraction across all test tiles is shown in Fig. 2e. In order to reconcile varying tile-wise median cloud fractions and identify a single tile-independent threshold, the median value – approximately 0.6
(indicated by the red asterisk) – was selected as the cloud fraction threshold. By setting the threshold at the median, we aim to





identify cloudy images while minimizing the risk of false positives (which could result in eliminating more images with a lower tile-independent threshold, although certain snowy pixels remain valuable, particularly for Canadian tiles) and false negatives (including more cloudy images that likely misclassify snow as cloud, with a higher tile-independent threshold, especially for the WUS tiles) in the cloud classification process. Therefore, Snow CCI fSCA images with an internal cloud fraction greater than 60% are removed.

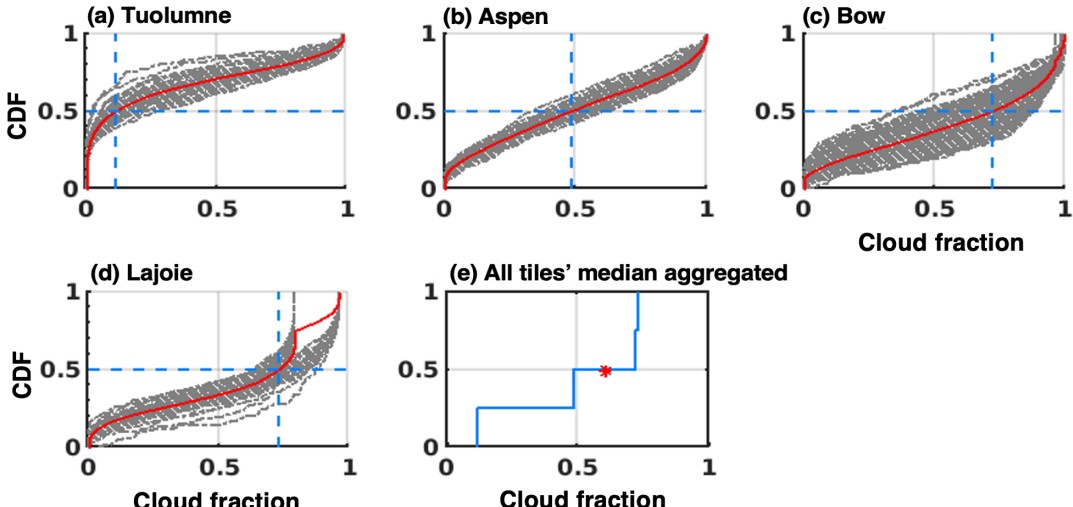

**Figure 2.** CDFs of tile-wise cloud fraction (aggregated by domains) for WY2000-2019 derived from the internal Snow CCI cloud mask (a-d). The grey dashed curves represent the CDF of cloud fraction for each WY, the red solid curve represents the CDF for the multi-year period (WY2000-2019). The intersection of blue dashed lines indicates the tile-wise median cloud fraction. The CDF of the median cloud fraction for all tiles, with a red asterisk representing the cloud fraction threshold (e).

### 2.3.2 Viewing geometry effects on the fSCA assimilation

A key difference between the Snow CCI fSCA and the Landsat fSCA is the sensor viewing geometry. The 16-day Landsat repeat frequency is due to its near-nadir viewing geometry. The Snow CCI fSCA product used in this study is derived based on data from the MODIS sensor, which has a wide swath (by scanning at different angles) to provide a daily revisit frequency. As a result, some daily images can have significant off-nadir viewing angles near the edge of the swath. This effect has been shown to lead to differences due to distorted irregular pixels at large zenith angles (Dozier et al., 2008), with negative impacts on the retrieval of fSCA especially in forested areas (Margulis et al., 2019). The primary result of this is that fSCA estimates should reflect less error when derived from near-nadir viewing than those pixels at significantly off-nadir viewing. Following Margulis et al., (2019), the error covariance of fSCA measurements that penalizes the off-nadir viewing geometry angles can be represented as



$$C_v^{Snow\ CCI}(\theta) = \frac{C_v^{Snow\ CCI}(\theta=0)}{w(\theta)} \tag{1}$$

where the measurement error covariance $C_v^{Snow\ CCI}(\theta)$ is a function of the MODIS sensor viewing angle $\theta$. The numerator $C_v^{Snow\ CCI}(\theta=0)$ is the error covariance at nadir, and $w(\theta)$ is a specified weighting function that is associated with non-nadir scan angles (more detailed explanations are in Dozier et al., 2008):

$$w(\theta) = \frac{p^2 cos\theta}{p_\| p_\perp} \tag{2}$$

where $p$ is the linear pixel dimension at nadir and $p_\|$, $p_\perp$ are the along-track and cross-track pixel dimensions at a non-nadir angle. The more off-nadir measurements (with more significant pixel elongation) have smaller $w$ values, and thus, larger measurement error covariances. We recognize that increasing the measurement error covariances does not correct any bias; biases induced by off-nadir effects may still introduce systematic errors in the posterior results. While these viewing geometry corrections may screen out highly problematic forested pixels that are significantly affected by the viewing geometry, it should be noted that Eq. (3) does not explicitly incorporate the underlying forest cover as a factor. Based on the assimilation of MODIS-based fSCA (Margulis et al., 2019), the error covariance at nadir (i.e., $C_v^{Snow\ CCI}(\theta=0)$) is specified as $\sim(15\%)^2$ for the MODIS-based Snow CCI fSCA. The relationship between the error covariance $C_v^{Snow\ CCI}(\theta)$ and $w(\theta)$ can be specified as

$$C_v^{Snow\ CCI}(\theta) = \frac{C_v^{Snow\ CCI}(\theta=0)}{w(\theta)} \approx \frac{(15\%)^2}{w(\theta)} \tag{3}$$

Note that the weighting function $w(\theta)$ varies within (0,1] by its definition. Following the screening method for the viewing geometry (Margulis et al., 2019), a threshold of $w(\theta)$ needs to be identified to exclude the measurements at pixels with significant distortions. The rationale behind selecting the threshold of $w(\theta)$ is the need to eliminate measurements with lower reliability, which could potentially introduce noise or inaccuracies into the assimilation. However, there is a trade-off between the threshold of $w(\theta)$ and the number of informative fSCA measurements for assimilation. If the threshold were set too high, there is a risk of discarding a substantial number of measurements, potentially leading to a loss of valuable fSCA information during the ablation season and limiting the effectiveness of assimilation.

To determine the threshold of $w(\theta)$, we show the impact of the $w(\theta)$ on the accuracy of the assimilated fSCA, where the accuracy (measurement error standard deviation) is given by $\sqrt{C_v^{Snow\ CCI}(\theta)}$, using $C_v^{Snow\ CCI}(\theta)$ from Eq. (3). A smaller $\sqrt{C_v^{Snow\ CCI}(\theta)}$ represents a more accurate Snow CCI fSCA. As shown in Fig. 3, the threshold of $w(\theta)$ $\sim$0.2 is a reasonable number to exclude less reliable measurements ($w(\theta)$ <0.2) with a sharp increase in the uncertainty at higher viewing angles that are unlikely to provide useful information in the assimilation step. As $w(\theta)$ approaches 1, the measurement error





approaches that of the Landsat fSCA. The measurement error of Landsat fSCA used in the Landsat reanalysis reference is
indicated by the red triangle (i.e., 0.1 at nadir where $w(\theta)$=0) in Fig. 3.

The boxplot in Fig. 4 displays the basin-average number of the assimilated fSCA per year for both Snow CCI and
Landsat, and specifically those that are informative. The informative fSCA measurements are those that contribute to the
posterior update and can be identified as those that occur when the prior ensemble spread of fSCA is greater than zero (most
typically during the ablation season). Additionally, the spatiotemporally averaged measurement error for each case is depicted
by the circle that is projected onto the right y axis. The Landsat error is fixed at 10%, while Snow CCI is 15% at a minimum,
but is closer to 20% on average when considering the viewing geometry. When comparing the two datasets, it is evident that
Landsat provides more accurate fSCA measurements with a lower measurement error at a lower repeat frequency, whereas
Snow CCI generally has a higher number of informative fSCA measurements but with higher measurement errors.

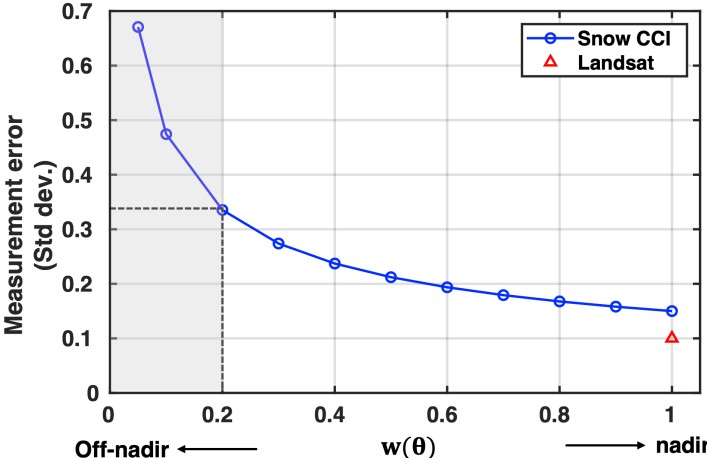

**Figure 3.** The impact of the $w(\theta)$ threshold on the accuracy, i.e. measurement error standard deviation, of Snow CCI fSCA
for assimilation. Areas below the threshold of $w = 0.2$ are excluded from the assimilation. The measurement error of Landsat
fSCA is represented by the red triangle (i.e., 0.1 at nadir).





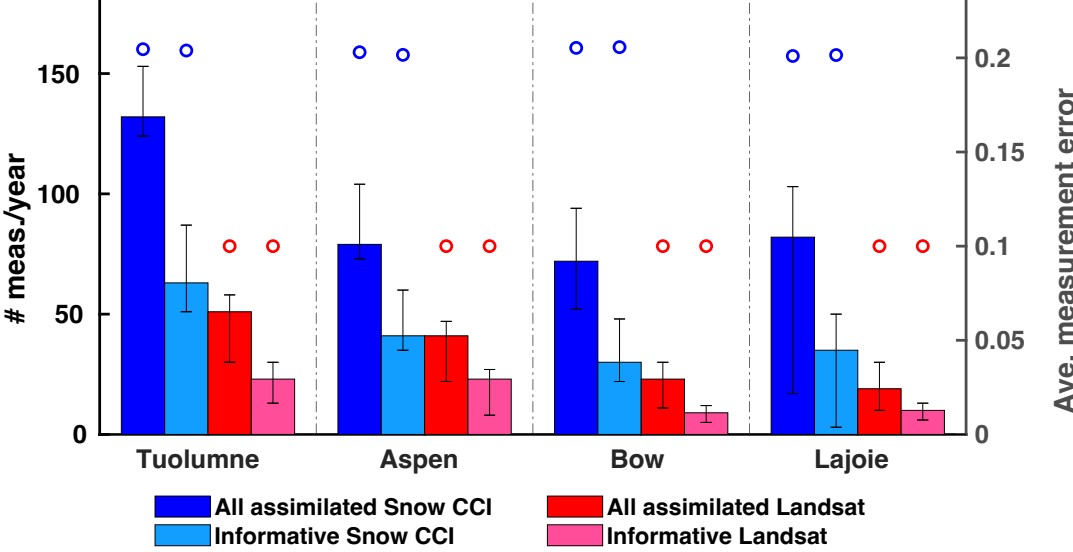

**Figure 4.** Left axis: Boxplot of the domain-average number of all assimilated and informative Snow CCI fSCA and Landsat fSCA measurements per year. Right axis: The spatiotemporally averaged measurement error for Snow CCI (in blue circles) and Landsat (in red circles).

Figure 5 summarizes the screening process for Snow CCI fSCA as described in Sect. 2.3.1 and 2.3.2. Figure 5a-c illustrates an example of the cloud screening with the threshold of 60% for the Snow CCI fSCA images. The erroneous cloudy images (e.g., Fig. 5a) are discarded as the cloud coverage of 76.5% is above the threshold, which is hypothesized to increase the likelihood of misclassifying cloud as snow. For the remaining images (e.g., Fig. 5b and 5c), pixel-wise clouds are screened out using the internal Snow CCI cloud mask before assimilation. The pixel-elongation effect caused by the off-nadir viewing geometry is illustrated by Fig. 5c, where the horizontal stretching pattern is clearly evident. Snow CCI fSCA measurements with pixel-wise $w(\theta)$ less than 0.2 from such images are further screened out. The screening process based on the viewing geometry angle is illustrated in Fig. 5d-f. For each pixel in the remaining images, Snow CCI fSCA with off-nadir geometry angles and/or masked by clouds are excluded before the SWE reanalysis.





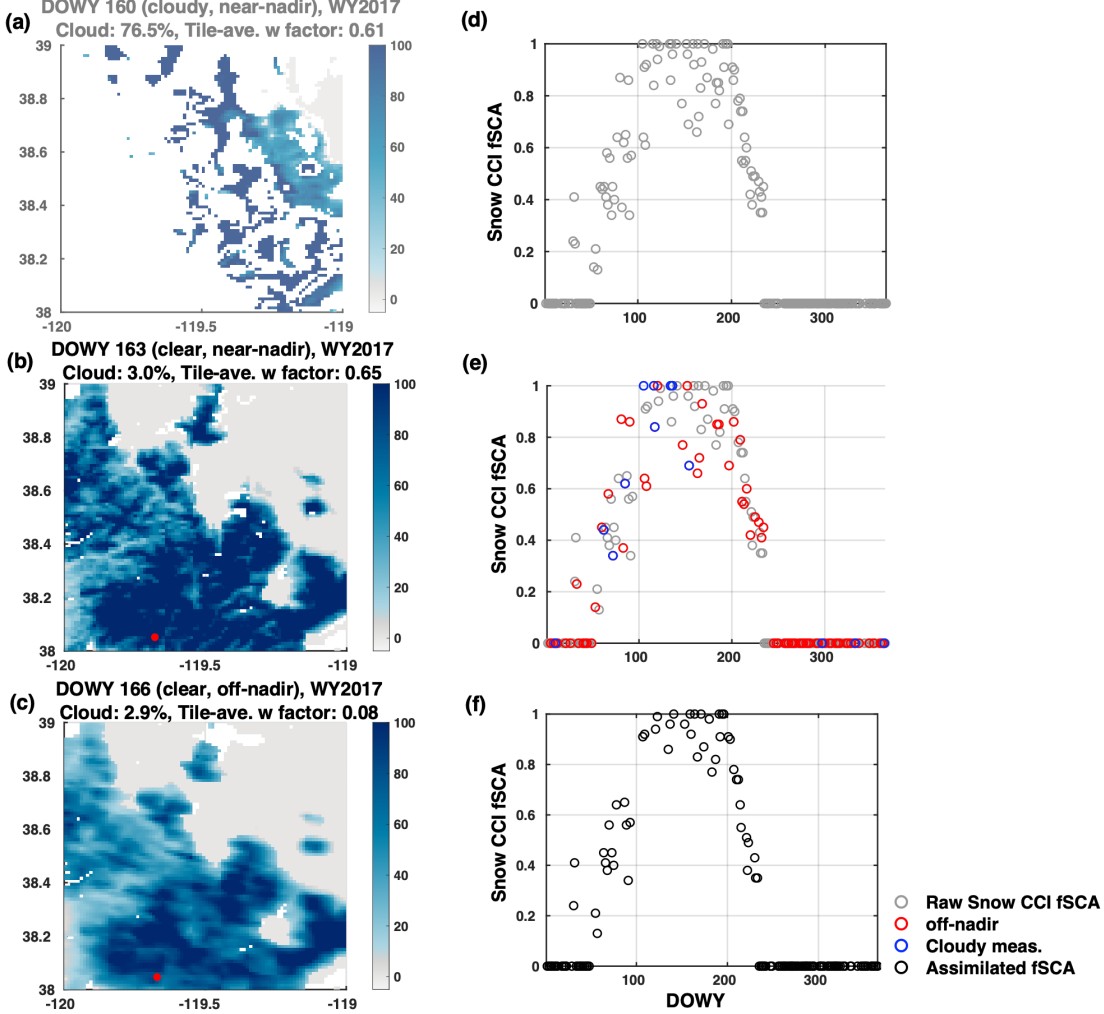

**Figure 5.** Illustration of cloud and off-nadir screening processes. The left panels (a-c) display the cloud screening of raw Snow
CCI fSCA images. The white color represents clouds, and grey color represents fSCA lower than 10%. Image (a) is discarded
completely due to cloud coverage above 60%. Image (b) has a domain-wide cloud fraction below the threshold, so only cloudy
pixels are screened out. Image (c) displays an off-nadir image on a clear day. Off-nadir measurements from image (c) are
excluded in the pixel-wise screening (e). The right panels (d-f) present the complete screening process of Snow CCI fSCA at
a sample pixel (indicated by a red dot on the left panel). The daily time series of Snow CCI fSCA show raw data (d), data to
be removed through pixel-wise cloud screening and off-nadir screening (e), and data to be assimilated (f).





## 2.4 Evaluation data and methods

### 2.4.1 Verification with independent datasets

The performance of the Snow CCI reanalysis estimates is evaluated through comparisons with independent in-situ
observations from automated snow pillows and manual snow courses as well as airborne lidar-based SWE estimates for the
WUS.  Each type of reference data has its strengths and weaknesses. Snow pillows (Beaumont, 1965) provide daily or even
hourly SWE estimates for a ~10 m$^2$ area, which may not be spatially representative at the 1 km$^2$ grid scale. However, the high
temporal frequency provided by automated snow pillows is helpful in capturing the seasonal evolution of SWE, important
within the BSRF framework which relies on a snow depletion curve (Liston, 2004).  Snow courses provide monthly and/or
biweekly SWE measurements. The measurements are averaged along a ~100-500 m long transect. It is expected that there are
discrepancies between the snow pillows and courses even if they are collocated, unless the underlying SWE is spatially
homogeneous. Despite the representativeness issue when comparing the in-situ measurements to 1 km$^2$ gridded estimates and
sampling a very small fraction of the test domains, we perform the verification in terms of the annual peak SWE and the
temporal correlation of daily SWE, where the correlation is insensitive to biases. Airborne Lidar data provides spatially
complete measurements of SD or estimates of SWE, but are available only for specific dates, typically near peak SWE.
Estimates of snow density are needed to go from SD to SWE (Painter et al., 2016).

In-situ SWE measurements are available from the Natural Resources Conservation Service (NRCS) via
https://wcc.sc.egov.usda.gov/reportGenerator/ for the WUS, and for Canadian domains, access is provided through the
Canadian historical Snow Water Equivalent (CanSWE) dataset (https://doi.org/10.5281/zenodo.7734616). Table 1 summarizes
the number of in-situ sites and years for each domain. SWE measurements are not available in Aspen basin and so only
comparison with airborne lidar estimates is possible over that domain.

The lidar-based ASO SWE estimates available at 50 m spatial resolution over domains in WUS are aggregated to the
Snow CCI SWE reanalysis resolution (0.01°) for the comparison. The ASO SWE estimates are only available for the domains
in the WUS for limited days in recent years (Table 2). To facilitate comparison, these measurements are interpolated to model
grids using the nearest neighbor interpolation method.

### 2.4.2 Comparison with the Landsat SWE reanalysis dataset

In the absence of spatiotemporally continuous reference SWE, we use a SWE reanalysis dataset produced by
assimilating cloud-free Landsat fSCA aggregated to a spatial resolution of 0.01°. This dataset is similar to that of Fang et al.
(2022) which was produced for the Western US at a spatial resolution of ~500 m (16 arcseconds) for 1985-2021. The specific
thresholds, uncertainty parameters, etc., are described in Fang et al. 2022 (Fang et al. 2022 Table 3). The performance of the
Landsat reanalysis SWE is well-understood over the western US but not over western Canada. We consider the performance
of the Landsat and Snow CCI reanalysis SWE against the independent reference data and relative to each other. Such





comparisons allow us to diagnose the performance of the Snow CCI reanalysis and understand to what extent the Snow CCI

fSCA can be used for the SWE reanalysis.

Table 1. Number of in-situ sites, site-years, and sources of SWE measurements used in this study

| Domains | Number of sites | Years | Source |
|---|---|---|---|
| **Automated snow pillows** | | | |
| Tuolumne, California | 4 | 2001-2019 | NRCS SNOTEL |
| Aspen, Colorado | 0 | N/A | NRCS SNOTEL |
| Bow River, Alberta | 5 | 2001-2019 | CanSWE v5. (Vionnet et al., 2021) |
| Lajoie, British Columbia | 2 | 2001-2019* one is only 2015-2019 | CanSWE v5. (Vionnet et al., 2021) |
| **Manual snow course** | | | |
| Tuolumne, California | 8 | 2001-2019 | CDEC snow courses |
| Aspen, Colorado | 0 | N/A | NRCS SNOTEL |
| Bow River, Alberta | 20 | 2001-2019 | CanSWE v5. (Vionnet et al., 2021) |
| Lajoie, British Columbia | 2 | 2001-2019 | CanSWE v5. (Vionnet et al., 2021) |

Table 2. Lidar-based measurement days over study domains in WUS (for SWE).

| ASO domains | Water Year | Day of Water Year |
|---|---|---|
| Tuolumne, California | 2015 | 185 |
| | 2016 | 184 |
| | 2017 | 183 |
| Aspen, Colorado | 2019 | 189 |


## 3 Results and Discussion

In this section, we first evaluate the performance of the reanalysis framework adapted from using Landsat fSCA to Snow CCI MODIS fSCA inputs within the well-validated WUS domains. By comparing with the previously validated Landsat reanalysis reference, we can analyze the impact of differences in fSCA datasets on the accuracy of the SWE estimation.

Subsequently, we extend the study regions to the western Canadian domains, which are forested and located at higher latitudes



than WUS. The reanalysis framework has not been applied to the western Canadian domains yet, and the performance has not been previously verified.

## 3.1 Application over previously studied WUS domains

### 3.1.1 Verification against in-situ SWE measurements

Posterior peak SWE estimates from both Snow CCI reanalysis and Landsat reanalysis are compared with in-situ measurements (snow pillows and courses) available in the Tuolumne watershed. In general, the Landsat reanalysis performs better than the Snow CCI reanalysis with a correlation of 0.91 and RMSD of 0.26 m, compared to a correlation of 0.47 and RMSD of 0.53 m. The scatterplot in Fig. 6 presents the comparison of the in-situ peak SWE against collocated posterior peak SWE estimated from the Snow CCI reanalysis. To incorporate the Landsat posterior SWE estimates into this comparison, we

use different colors to represent absolute relative differences in basin-average peak SWE estimated from Snow CCI relative to Landsat. These values are computed as absolute differences in basin-average peak SWE, normalized by basin-average peak SWE from the Landsat reference. Lower values of absolute relative differences indicate years when Snow CCI posterior SWE is more similar to Landsat posterior SWE on a basin scale, and vice versa. For example, the Snow CCI estimates closely match the in-situ measurements when the absolute relative differences are within the range of 0-0.1 (in red). Conversely, the estimates

are more scattered when the Snow CCI estimates diverge significantly from the Landsat reference (e.g., greater than 0.5 in grey). The right panel in Fig. 6 displays the statistics (i.e., correlation and RMSD) of the verification against in-situ peak SWE. The statistics are computed for each range bin, using data points represented in different colors. The Snow CCI reanalysis performs well in years when the basin-average peak SWE match that of the Landsat reference and the relative differences are lower than 0.3. In such cases, correlation values are greater than 0.8, and the RMSD are lower than 0.5 m. However, other

years with divergence in basin-average peak SWE between Snow CCI and Landsat exhibit lower correlation and higher RMSD values relative to the in-situ measurements. The reason for discrepancies between Landsat- and Snow CCI-derived SWE is discussed below in Sect. 3.1.3.





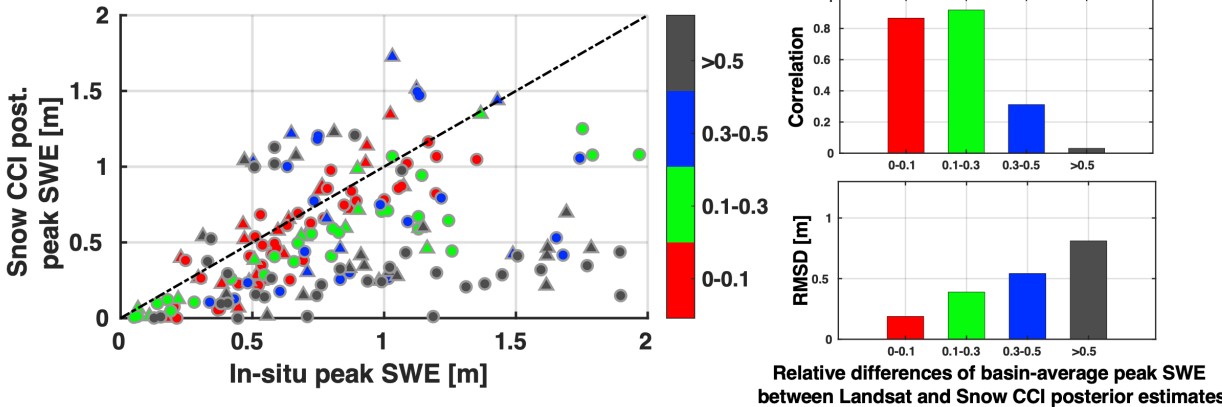

**Figure 6.** Scatterplot of Snow CCI posterior peak SWE vs. in-situ peak SWE at snow pillow sites (triangles) and snow course sites (circles) the Tuolumne domain. The color represents the relative differences of basin-average peak SWE relative to the Landsat posterior SWE reference. Bar plots show variations in correlation and RMSD with in-situ measurements across different ranges of relative peak SWE differences relative to the Landsat reference.

Figure 7 displays the temporal (daily) SWE comparison at snow pillow sites in the Tuolumne domain. The locations of snow pillow sites are shown in Fig. 1. The correlation is computed by comparing Snow CCI and Landsat posterior daily SWE against in-situ daily SWE greater than 1 cm. In the Tuolumne domain, posterior daily SWE at snow pillow sites have high correlations against in situ SWE. The average correlation values are 0.89 and 0.95 for Snow CCI and Landsat reanalysis, respectively. The bottom panel depicts differences in the correlation relative to the prior daily SWE comparison. Specifically, cases in blue colors and highlighted by the black boxes represent site-years where the reanalysis improves the correlation of daily SWE. For the Snow CCI reanalysis, 30 out of 59 site-years show improvements in the correlation, while the assimilation of Landsat fSCA improves the correlation for 48 out of 59 site-years. For some years where there is degradation in the correlation after assimilating Snow CCI fSCA, such as 2003, 2006, 2011, and 2019, Snow CCI fSCA is likely negatively biased over the domain during the ablation season (Fig. 10). Such biases in fSCA could cause significant shifts in the time of peak SWE and the snowmelt season, which degrade the correlation of the daily SWE compared to in-situ measurements.





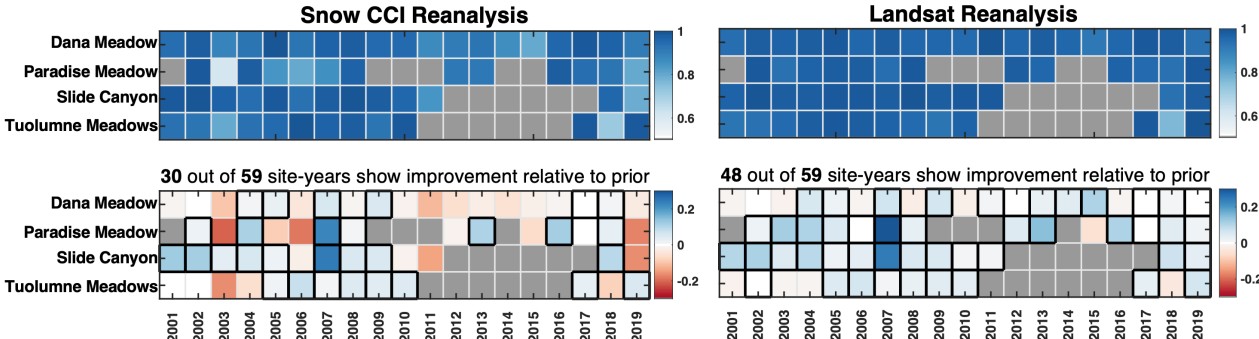

**Figure 7.** Temporal correlation of Snow CCI (upper left) and Landsat (upper right) posterior daily SWE vs. snow pillow daily SWE measurements at snow pillow sites in the Tuolumne domain. The bottom panels display differences between the posterior correlation and the prior correlation (posterior R – prior R). Station-years with improvements are in the black boxes. Cases where snow pillow measurements are incomplete and/or annual peak SWE values were lower than 2 cm are greyed out.

### 3.1.2 Verification against ASO SWE

For the Tuolumne domain, the assimilation of Snow CCI fSCA significantly improves the correlation relative to the prior SWE on the days near April 1st compared to the ASO SWE (Table 3). The posterior Snow CCI SWE is highly correlated with ASO SWE, with correlation values ranging from 0.75 to 0.87, while the prior correlation values range from 0.48 to 0.54. The most significant improvement in the correlation occurs in WY 2017 (Fig. 8) where the RMSD decreases from 0.6 m to 0.46 m. The assimilation of the aggregated Landsat fSCA shows a larger improvement in all WYs in terms of the correlation and RMSD (Table 3). The posterior Landsat SWE exhibits high correlation values ranging from 0.83 to 0.92, comparable to the statistics in previous work (refer to Table 6 in Fang et al., 2022). Figure 8 shows that in Tuolumne, the prior SWE has positive biases over less snowy areas at lower elevations and negative biases over more snowy areas at higher elevations. After assimilating the aggregated Landsat fSCA, the systematic errors are reduced, with random errors more dispersed across the domain. When assimilating Snow CCI fSCA, negative biases exist in posterior SWE across most of the domain.

For the Aspen in Colorado (Table 3), the correlation of the posterior SWE in WY 2019 is comparable to the value of the prior SWE, while the RMSD decreases from 0.45 m to 0.33 m (for Snow CCI) and 0.21 m (for Landsat). The correlation values are lower than the values seen in the Tuolumne domain. This could be because the snow albedo uncertainty is not well characterized in Colorado (Fang et al., 2022) and wildfires in recent years (e.g., 2018) enhance variations in the albedo and surface brightness. The spatial map (Fig. 8) shows that the prior SWE is higher than the ASO SWE across Aspen. The assimilation of remotely sensed fSCA lowers the SWE estimates, with Snow CCI exhibiting more negative biases than the Landsat reference.



Table 3. SWE comparison statistics between ASO SWE estimates against prior and posterior snow reanalysis SWE on ASO measurement days (Day of Water Year, DOWY) closest to April 1st.

| ASO basin | Year | DOWY | Correlation | | | RMSD [m] | | |
|---|---|---|---|---|---|---|---|---|
| | | | Prior | Landsat Posterior | Snow CCI Posterior | Prior | Landsat Posterior | Snow CCI Posterior |
| Tuolumne | 2015 | 185 | 0.48 | 0.83 | 0.75 | 0.07 | 0.047 | 0.055 |
| | 2016 | 183 | 0.64 | 0.89 | 0.85 | 0.37 | 0.22 | 0.32 |
| | 2017 | 183 | 0.54 | 0.92 | 0.87 | 0.6 | 0.27 | 0.46 |
| Aspen | 2019 | 189 | 0.51 | 0.53 | 0.54 | 0.45 | 0.21 | 0.33 |

370

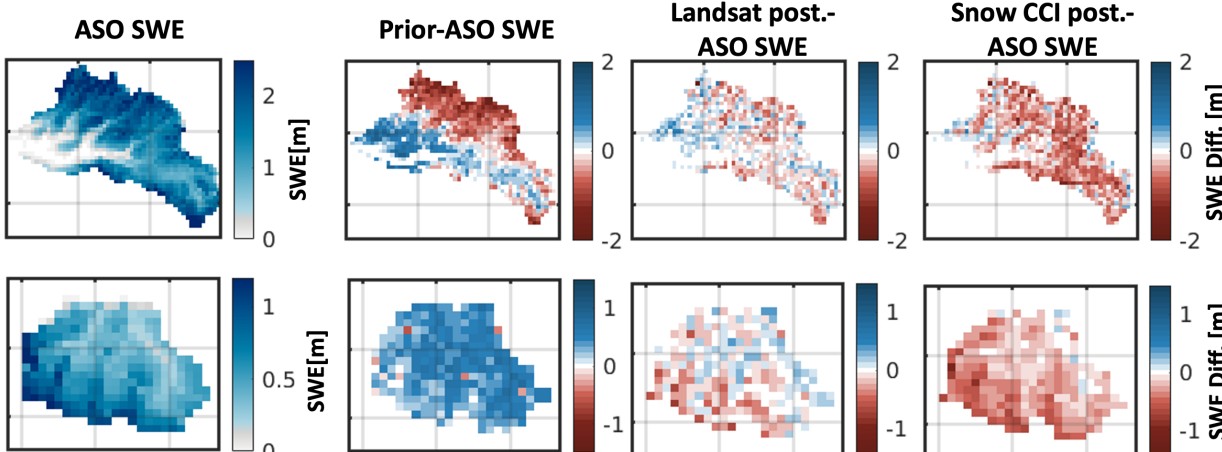

**Figure 8.** Comparison of ASO SWE with prior and posterior SWE at WUS ASO sites: Tuolumne basin on April 1st, WY2017 and Aspen on April 8th, WY2019.

375 **3.1.3 Comparison to Landsat-based reanalysis**

**3.1.3.1 Contributing factors to fSCA differences**

The performance of SWE reanalysis over mountain areas is primarily affected by the accuracy of the assimilated fSCA and the prior estimates. Since the prior estimates are identical for the Landsat and Snow CCI reanalysis, any differences in the posterior are due to the fSCA. The annual and interannual comparison of fSCA observations from Snow CCI against

380 Landsat provide insight on the impact of fSCA differences on the accuracy of SWE estimation.

A key difference between the two fSCA products is resolution: the raw nadir resolution of Landsat (~30 m) is significantly higher than MODIS (~500 m), which can improve its ability to resolve spatial patterns and see through forest





gaps. Snow CCI tends to underestimate fSCA in forested areas (Fig. S1). Additionally, the broader range of viewing angles observed by the MODIS sensor has a "smearing" effect that elongates pixels when viewed at higher zenith angles (Dozier et al. 2008; Fig. 5). Beyond considerations of resolution and viewing angle, different retrieval algorithms are used to derive Snow CCI and Landsat fSCA, which can lead to both systematic (biased) and random differences. Images of Landsat fSCA used in this study are retrieved via a spectral end-member unmixing approach (Cortés et al., 2014). Snow CCI fSCA images are derived using the SCAmod algorithm (Metsämäki et al., 2012, 2015), a semi-empirical method retrieving fSCA using observed reflectance with pre-determined parameters. Recognized issues associated with SCAmod include the upper/lower bounds of fSCA, which can exceed 0-1 when the observed reflectance is higher/lower than the limits allowed by the semi-empirical model (Metsämäki et al. 2015). Although Snow CCI fSCA constrains the bounds to 0-1, biases could exist when the retrieved fSCA is near the bounds. For example, Snow CCI fSCA tends to overestimate Landsat fSCA over bare soil areas with fSCA values near 100% (Fig. S1). The SCAmod algorithm also applies a temperature threshold of 288 K to mitigate the misclassification of snow cover caused by reflective non-snow targets. This temperature threshold has a tendency to remove low fSCA values (Riggs & Hall, 2012). Snow CCI fSCA tends to underestimate compared to Landsat fSCA during the late ablation season when snow cover depletes (Fig. S1).

### 3.1.3.2 Long-term SWE climatology

The spatial pattern of climatological peak SWE from the Snow CCI reanalysis is similar to the Landsat-based reference with correlation values of 0.89 and 0.86 for Tuolumne and Aspen, respectively (Table 4). Both spatial maps of peak SWE and time series of the seasonal cycle (Fig. 9) show that Snow CCI underestimates SWE over the Tuolumne domain, and is closer to the Landsat reference over Aspen. The mean difference of peak SWE is -0.16 m for the Tuolumne and -0.03 m for the Aspen, while the corresponding RMSD is 0.21 m and 0.1 m (Table 4). The impact of forest cover and aspect on the accuracy of Snow CCI SWE estimates is illustrated in Fig. 9c. In both ASO domains, the relative difference of posterior SWE increases with the forest cover fraction. This is likely because the Snow CCI fSCA has a coarser spatial resolution and cannot see through tree gaps within forested pixels as well as Landsat can. Additionally, the relative difference of posterior SWE is more negative in areas facing north and less negative (even positive) in areas facing south/east in both domains. South facing slopes tend to receive more shortwave radiation, leading to more reflectance compared to the north facing slopes. Note that Snow CCI fSCA is retrieved based on the SCAmod algorithm, which uses the observed reflectance data along with predetermined reflectances of snow, snow-free ground, and forest canopy via a semi-empirical reflectance model-based method (Metsämäki et al., 2015). It is possible that the prevailing snow reflectance on south facing slopes is higher than that applied in the SCAmod algorithm, resulting in higher values of Snow CCI fSCA than Landsat (Metsämäki et al., 2015). This result highlights a need for further work on the fSCA retrieval algorithm in relation to terrain aspect.



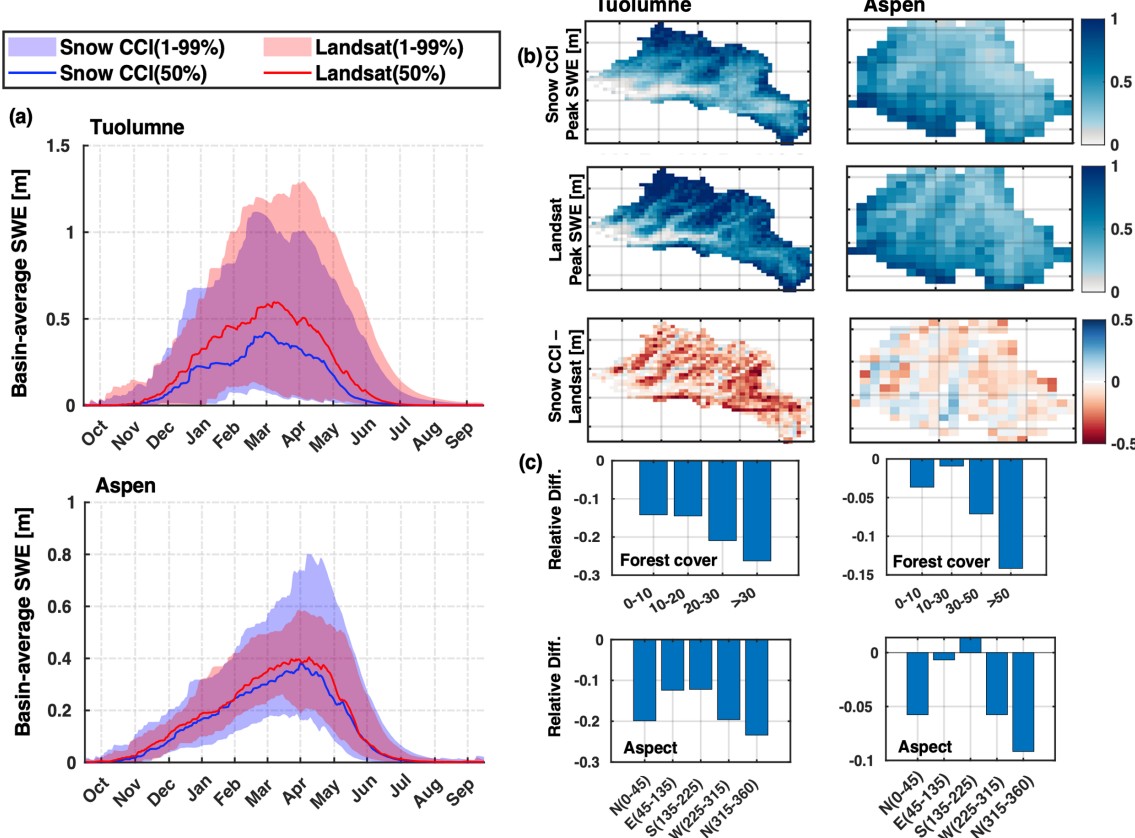

**Figure 9.** (a) Average seasonal cycle of basin-average posterior SWE from WY 2001 to WY 2019. The 19-year averages are displayed in solid lines, while the shaded regions represent the full range across WYs. (b) Spatial maps of the 19-year peak SWE and differences relative to the Landsat estimate. (c) Bar plots of relative differences (i.e., (Snow CCI - Landsat)/Landsat) as functions of forest cover and aspect.

Table 4. Comparison statistics for the spatial maps of 19-year median peak SWE between Snow CCI and Landsat posterior estimates.

| Domain | Correlation | Mean Difference [m] | RMSD [m] |
|---|---|---|---|
| Tuolumne | 0.89 | -0.16 | 0.21 |
| Aspen | 0.86 | -0.03 | 0.10 |

### 3.1.3.3 Interannual variability of SWE

The interannual variability of Snow CCI posterior SWE is similar to the Landsat-based estimates in most WYs on the basin-average scale (Fig. 10a). The basin-average differences in Snow CCI posterior fSCA estimates compared to the Landsat



reference are averaged for each month from February to September (Fig. 10b displays the Tuolumne and Aspen domains as examples). The differences in the basin-average peak SWE for all WYs are significantly correlated with the average fSCA differences during the ablation season across both domains, indicated by a correlation of 0.93 (Fig. 10c). The snowmelt timing suggested by fSCA datasets also influence the accuracy of SWE estimation. The differences in fSCA melt-out month are

correlated with the differences in peak SWE with R=0.97 (Fig. 10c).

For illustration purposes, two sample WYs with comparable performance are highlighted with blue boxes (i.e., 2002 and 2010 for the Tuolumne, 2006 and 2012 for the Aspen) while two typical WYs of differing performance are highlighted with magenta boxes (i.e., 2006 and 2011 for the Tuolumne, 2011 and 2018 for the Aspen). The fSCA values during the ablation season and snowmelt timing estimated from Snow CCI reanalysis match the Landsat-based estimates for the comparable WYs.

In contrast, significant biases exist in the snowmelt duration and fSCA values for WYs in blue boxes. In the Aspen domain, peak SWE generally occurs in April, with February and March receiving substantial snowfall. While the seasonal cycle of SWE is comparable to the Landsat reference (Fig. 9a), WY 2018 is a case of poor performance, but with significant positive biases in posteriori SWE. Correspondingly, Snow CCI predicts an abnormally long snowmelt season (April to September) characterized by significant positive biases in fSCA compared to the Landsat reference, particularly before July. The summer

of WY 2018 was particularly warm and dry with significant wildfires in Colorado. Positive biases in Snow CCI fSCA over the ablation season of WY 2018 may be related to non-identified clouds and warm bright land surface. This is a unique case where Snow CCI exhibits low quality in the Aspen domain, and the specified measurement error cannot correct biases in fSCA.



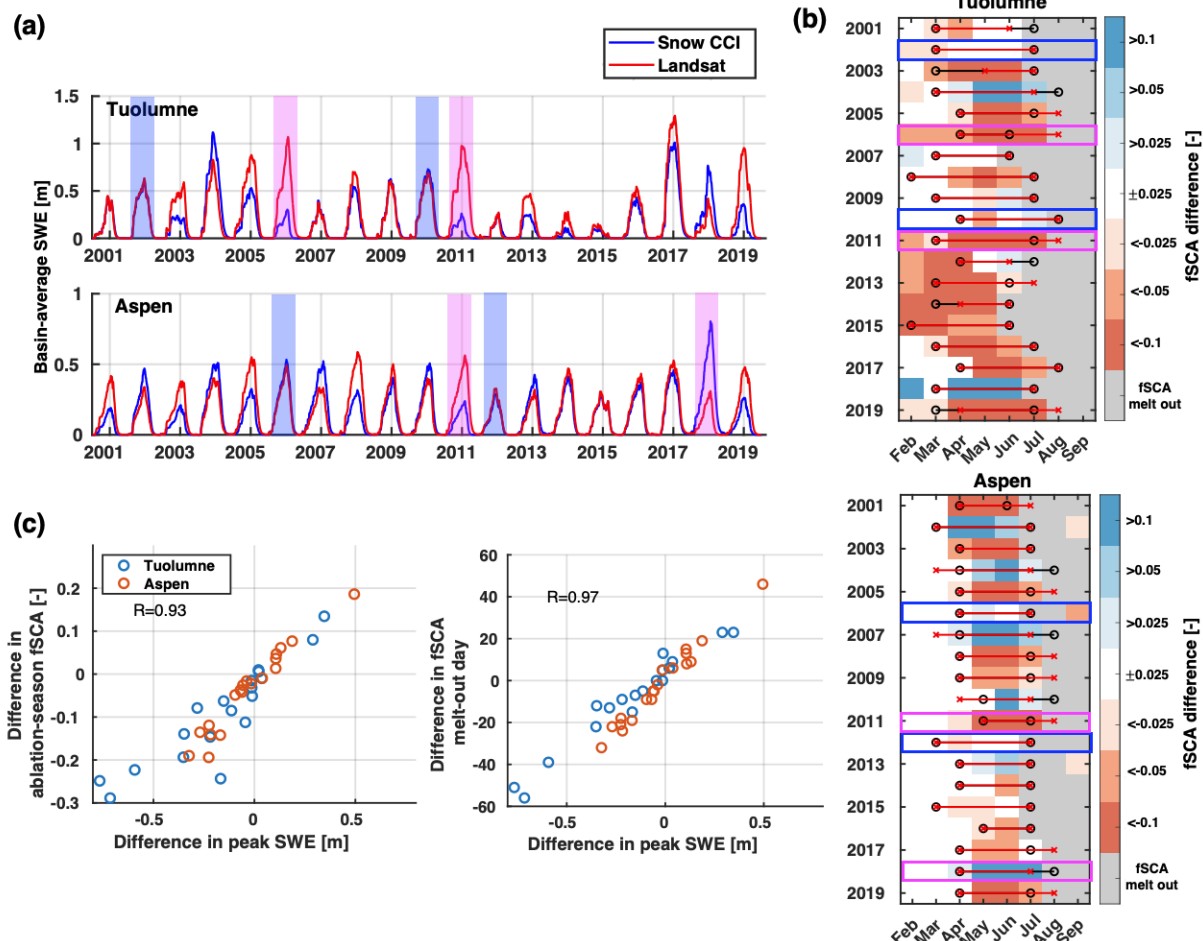

**Figure 10.** (a) Daily time series of basin-average posterior SWE from Snow CCI and Landsat reanalysis. (b) Monthly basin-average posterior fSCA differences (i.e., Snow CCI fSCA-Landsat fSCA). Months when both datasets indicate snow cover melt-out are greyed out. The black circles indicate the snowmelt season estimated from the Snow CCI reanalysis (peak SWE time and the time when fSCA melt-out), and the red crosses represent the snowmelt season from the Landsat reanalysis. (c) Scatter plots of differences in peak SWE vs. mean differences in the ablation-season fSCA, and vs. differences in fSCA melt-out day.

The interannual differences in the basin-average posterior SWE are also associated with the number of high-quality fSCA measurements in the ablation season. Due to the limited number of coincident days when both Snow CCI and Landsat fSCA are available, the quality of the Snow CCI fSCA at each pixel is identified through a comparison with the posterior fSCA curve constrained by the Landsat fSCA. The Snow CCI fSCA measurements that are more than $\pm 0.15$ (i.e., the measurement error of Snow CCI fSCA at the nadir angle) from the Landsat posterior fSCA curve are excluded. The number of Snow CCI





fSCA measurements varies across the years as does the basin-average ratio of the number of high-quality vs. total number of measurements in the ablation season (Fig. 11). This ratio is negatively correlated with the absolute relative difference in basin-average peak SWE compared to the Landsat reference. In general, biases in the basin-average peak SWE tend to decrease as the availability of high-quality Snow CCI fSCA data in the ablation season increases. Such a relationship is significant for the

Tuolumne domain (correlation of -0.61) but less significant for the Aspen domain (correlation of -0.28, -0.32 when anomalous WY 2018 is excluded). In sample cases of good performance (as illustrated in Fig. 10), the fraction of high-quality Snow CCI fSCA measurements is typically higher than 0.4, whereas the sample cases of poor performance have a fraction as low as 0.2. The abnormal WY 2018 in the Aspen domain, highlighted by the grey circle in the scatter plot, is characterized by fewer than 30% of all fSCA measurements being good quality (Fig. 11) and by significant positive biases in Snow CCI fSCA during the

ablation season (Fig. 10). Together, these factors contribute to a significant relative difference (>1.5) in the basin-wide peak SWE compared to Landsat.

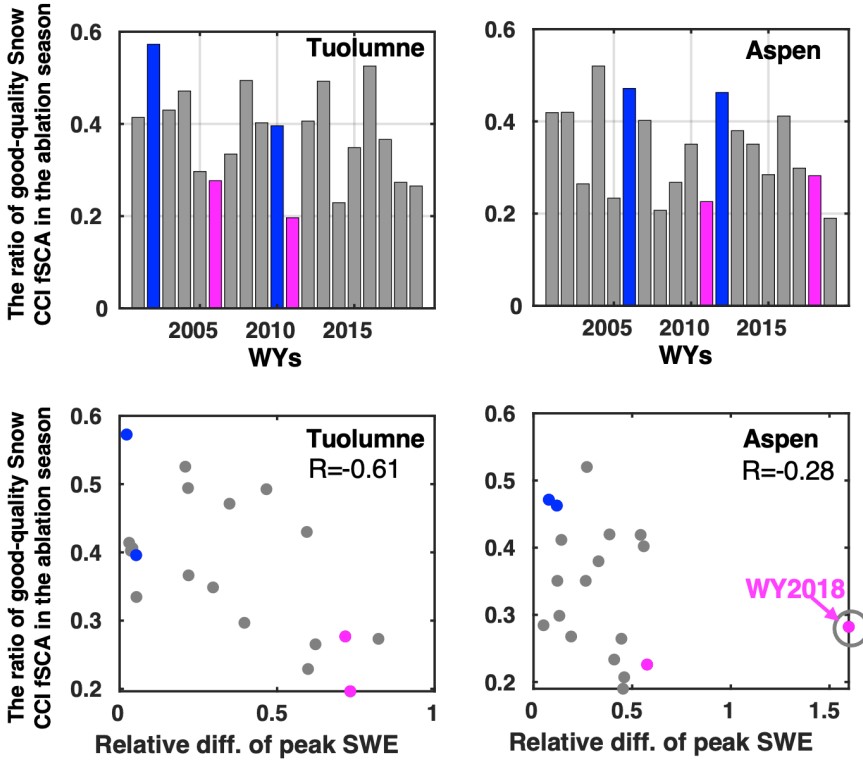

**Figure 11.** Bar plots (top) of the ratio of high-quality Snow CCI fSCA in the ablation season relative to Landsat fSCA. Scatter plots (bottom) of the ratio of high-quality Snow CCI fSCA in the ablation season vs. relative differences of basin-average peak

SWE. Cases of good performance vs. poor performance are highlighted in blue and magenta, respectively, as illustrated in Fig. 10.





## 3.2 Application over the western Canadian domains

### 3.2.1 Verification against in-situ SWE measurements

The performance of Snow CCI reanalysis is evaluated through the comparison of peak SWE against in-situ sites in
the Bow and Lajoie domains. The Landsat reference is validated in parallel since, unlike the WUS example, the SWE reanalysis
has not been applied to Canadian domains in previous work. Table 5 summarizes the statistics of the comparison of peak SWE
against in-situ measurements for both the Snow CCI and Landsat reanalysis. The Landsat reference performs well in both
basins with high correlation values of 0.61 and 0.75 and low RMSD of 0.2 m and 0.21 m for Bow and Lajoie, respectively.
We acknowledge that the number of site-years is limited in Lajoie domain, and the verification is constrained to only part of
the domains. The posterior SWE estimates from the Landsat reference have better accuracy than the Snow CCI reanalysis in
both domains relative to available in-situ measurements. Therefore, similar to the WUS basins, Landsat reanalysis will serve
as a comparison reference.

Table 5. Comparison statistics for the 19-year peak SWE between reanalysis posterior estimates and in-situ measurements.

| Domain | # site-years | Correlation | | RMSD [m] | |
|---|---|---|---|---|---|
| | | Landsat Post. | Snow CCI Post. | Landsat Post. | Snow CCI Post. |
| Bow | 653 | 0.61 | 0.46 | 0.20 | 0.24 |
| Lajoie | 60 | 0.75 | 0.16 | 0.21 | 0.49 |


Figure 12 displays the comparison of Snow CCI posterior peak SWE against in-situ peak SWE, where the colors
represent the relative differences in basin-average peak SWE compared to the Landsat reference. For both domains, the
correlation between posterior Snow CCI peak SWE and in-situ peak SWE can be as high as 0.6 for the relative difference bin
of 0-0.2 and can be as low as 0.4 and negative for the relative difference bin of >0.4. The RMSD in the peak SWE is higher
when the basin-wide relative differences are larger at the basin scale. For the Lajoie domain, the number of snow pillows and
snow courses is limited. The performance of Snow CCI reanalysis is especially poor at the low-elevation densely forested
Green Mountain site (Fig. 12 transparent colors), where Snow CCI SWE estimates are significantly lower than in-situ SWE,
which significantly degrades the statistics.



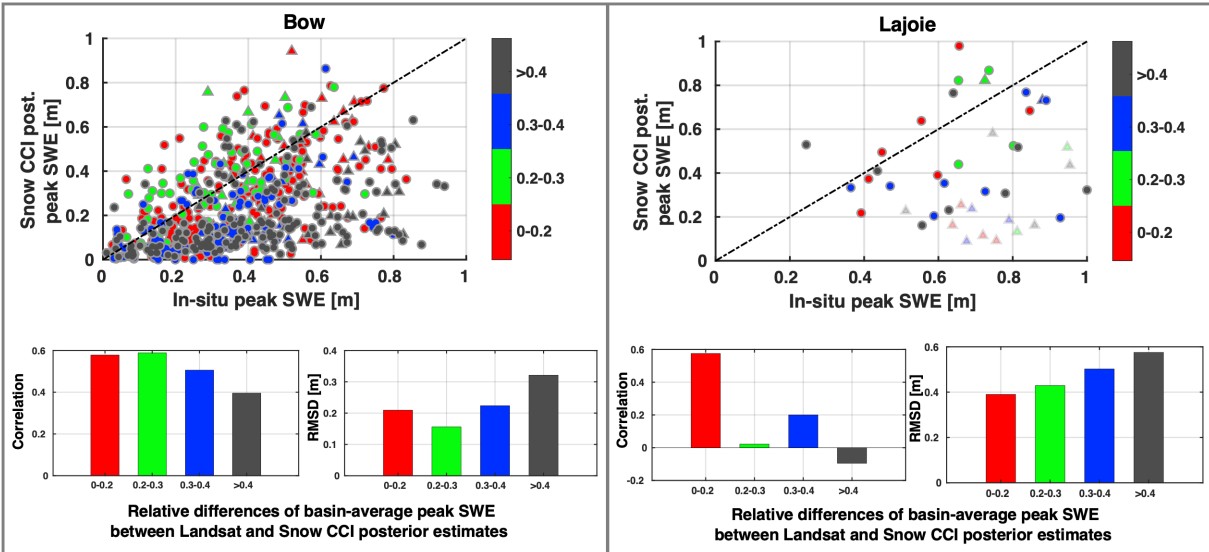

**Figure 12.** Same as Fig. 6, but for Bow and Lajoie domains. Transparent colors for Lajoie domain indicate the performance at the Green Mountain site described in the text.

The temporal (daily) SWE comparison (similar to Sec. 3.1.1) is performed at snow pillow sites in Bow and Lajoie domains over all WYs (Fig. 13). The average correlation between in-situ and posterior daily SWE is 0.92 and 0.94 for the Snow CCI and Landsat, respectively. The temporal (daily) correlation is much stronger than that of peak SWE. This is likely due to the strong seasonal cycle of SWE and the fact that the temporal comparison is unaffected by systematic biases. The assimilation of Snow CCI fSCA (Landsat) improves the correlation in 61 (69) out of 109 site-years compared to the correlation of prior estimates.

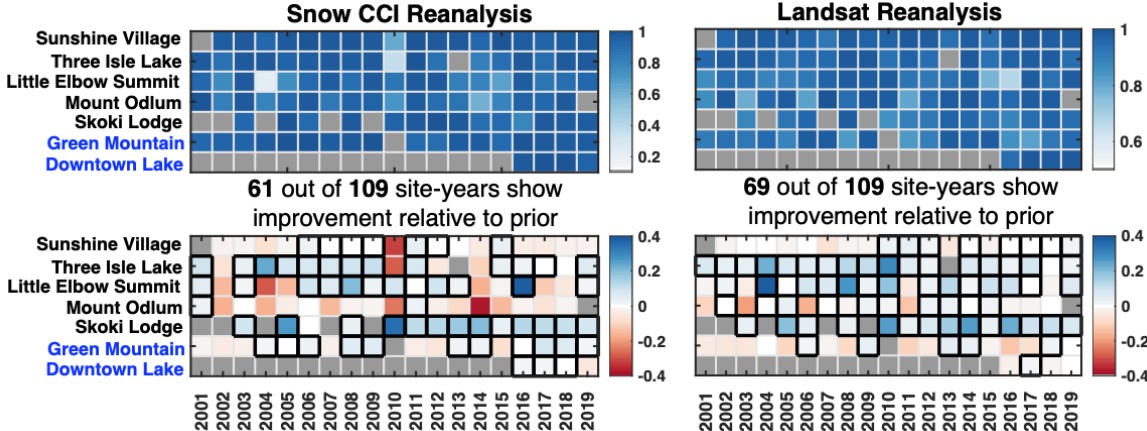

**Figure 13.** Same as Fig. 7, but for Bow and Lajoie domains with snow pillows names in black and blue, respectively.





Typical fSCA and SWE time series at sample in-situ sites in the Bow and Lajoie domains is displayed in Fig. S2. In selected samples, Snow CCI fSCA is generally lower than Landsat fSCA at the densely forested site (Mount Odlum), leading to an earlier snowmelt and lower SWE compared to both snow pillow measurements and Landsat reanalysis estimates. In contrast, Snow CCI fSCA is saturated and higher than Landsat fSCA at the bare soil site (Downton Lake), resulting in higher SWE and a later snowmelt season. Nevertheless, Snow CCI fSCA can provide benefits when the availability of Landsat fSCA is limited at the moderately forested site (Sunshine Village).

### 3.2.2 Landsat comparison

The comparison of the seasonal cycle of the basin-average SWE for the Bow and Lajoie domains is shown in Fig. 14a. The median seasonal cycle of Snow CCI posterior SWE generally matches the Landsat reanalysis with slightly negative differences. The full range of Snow CCI posterior SWE for the Lajoie domain is larger than that of Landsat posterior SWE, indicating a higher interannual variability. The spatial pattern of long-term peak SWE climatology from Snow CCI reanalysis is compared with the Landsat reference (Fig. 14b and Table 6). Snow CCI and Landsat posterior peak SWE have similar spatial patterns with correlations of 0.82 and 0.8 for Bow and Lajoie domains, respectively. Snow CCI and Landsat have comparable peak SWE for the Bow domain (mean difference of -0.04 m and RMSD of 0.09 m), whereas more significant differences exist in the Lajoie with a mean difference of -0.13 m and RMSD of 0.34 m.

To investigate the impact of forest cover fraction and aspect on relative differences in peak SWE identified in the WUS example (3.1.3.2), we similarly bin the relative differences in long-term peak SWE from Snow CCI vs. Landsat according to forest cover and aspect (Fig. 14c). In both domains, relative differences generally become more negative as forest cover increases, Bow forest cover 0-10% excepted. Regions with south-facing slopes tend to exhibit less negative and positive relative differences in peak SWE. The largest (and most negative) relative differences in peak SWE are found on north-facing slopes for both domains, indicating that Snow CCI may underestimate fSCA over lower illumination areas. Another hypothesis, as discussed in 3.1.3.2, is that small snow-covered areas (low fSCA) may still exist on north-facing slopes during warm season that are captured by the Landsat algorithm but are set to zero in Snow CCI by the temperature threshold screening (Metsämäki 2015).




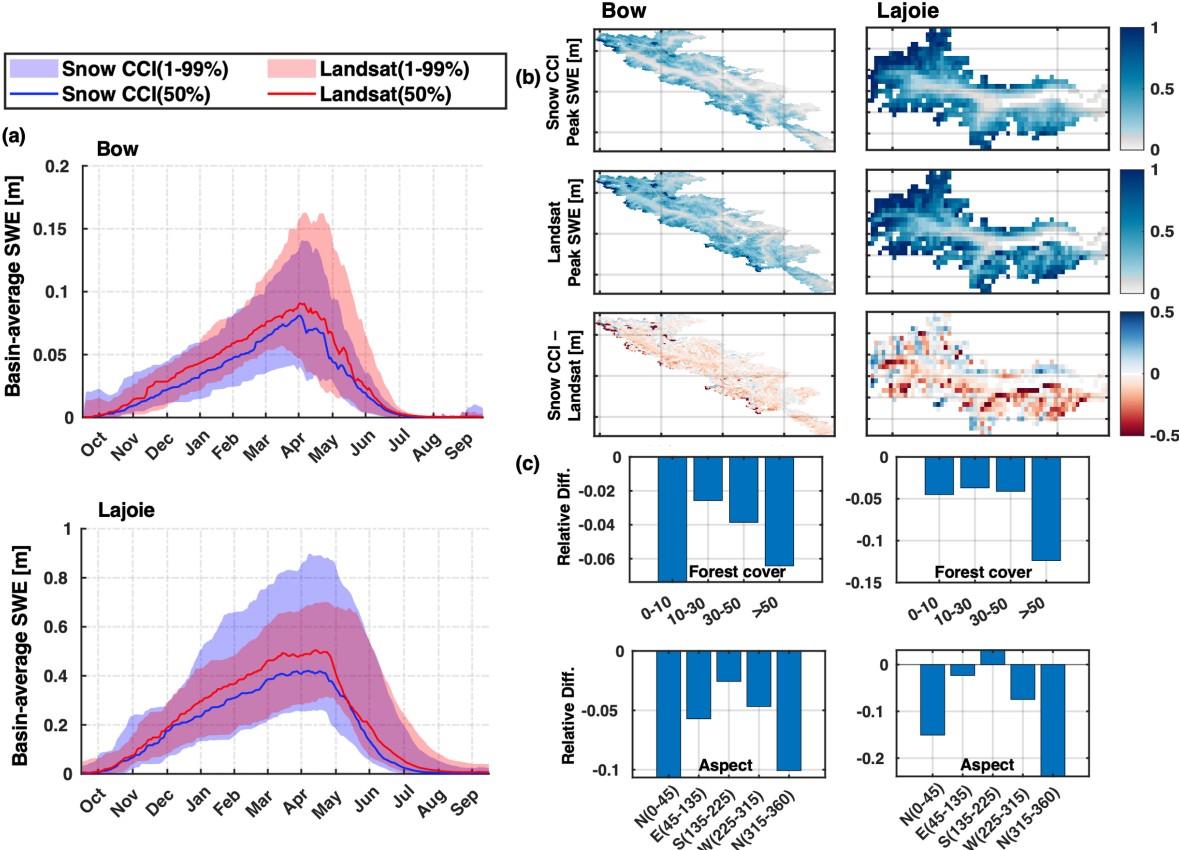

**Figure 14.** Same as Fig. 9, but for Bow and Lajoie domains.

535 Table 6. Comparison statistics for the long-term median peak SWE between Snow CCI and Landsat posterior estimates.

| Domain | Correlation | Mean Difference [m] | RMSD [m] |
|---|---|---|---|
| Bow | 0.82 | -0.04 | 0.09 |
| Lajoie | 0.80 | -0.13 | 0.34 |

The interannual variability of basin-average SWE from Snow CCI vs. Landsat reanalysis is displayed in Fig. 15a (Sect. 3.1.3.3). Snow CCI reanalysis generally yields comparable basin-average SWE relative to the Landsat results in most WYs. As was demonstrated in the WUS examples, significant fSCA differences in the ablation period lead to significant
540 differences in basin-average peak SWE. This is also true in the Canadian test cases (Fig. 15c) as indicated by the correlation value of 0.84. It is also apparent that the earlier fSCA melt-out timing corresponds to the lower peak SWE and vice versa (Fig. 15c), and the correlation value is 0.93.





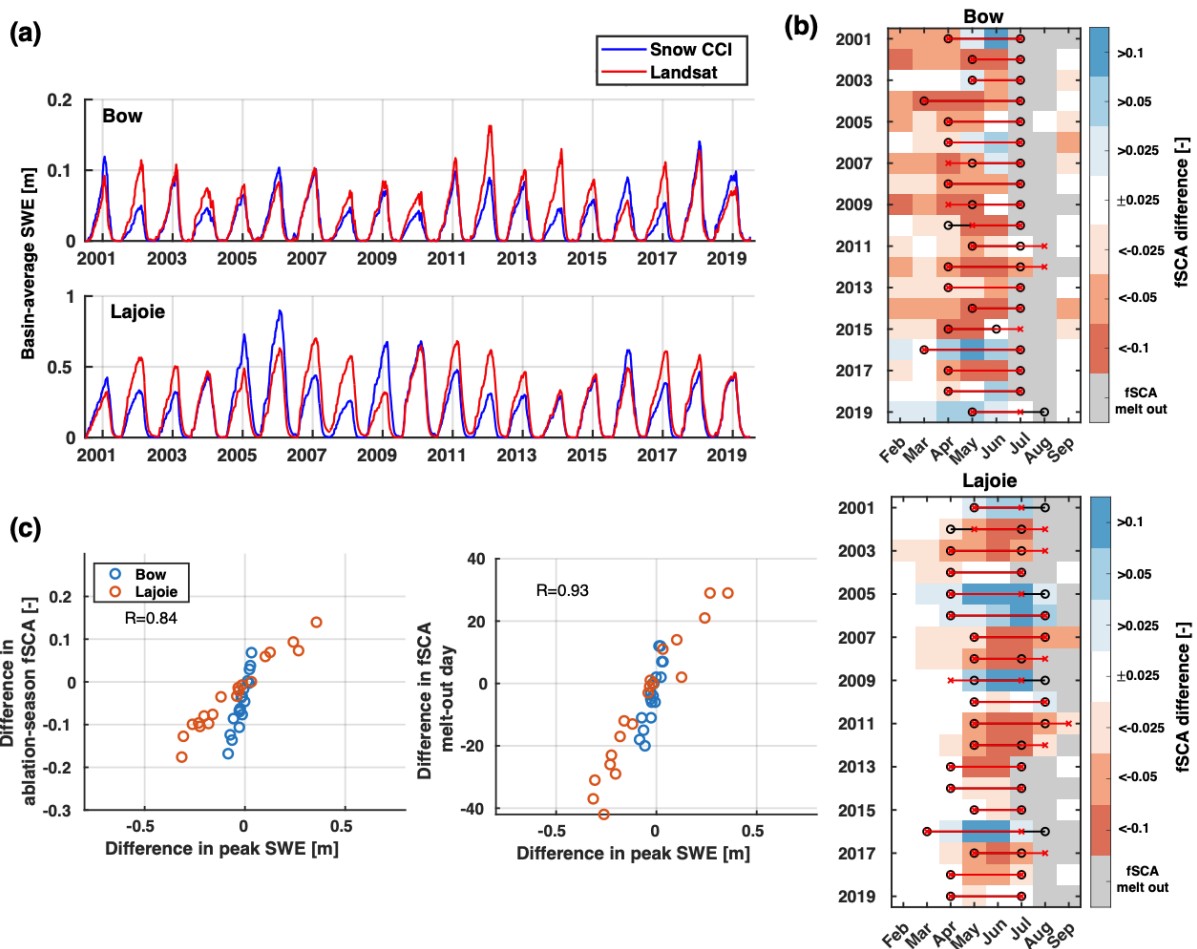

**Figure 15.** Same as Fig. 10, but for Bow and Lajoie domains.

## 4 Conclusions

This study explored the potential for using the Snow CCI fSCA Climate Data Record (CDR) for mountain SWE estimation using a Bayesian SWE reanalysis framework. Four application domains spanning the WUS and western Canada with various physiographic and climatological conditions are used in the evaluation of the SWE estimation performance. Since Snow CCI fSCA is retrieved based on reflectance observations from the MODIS sensor, we applied the measurement error (standard deviation of 15%) used for other MODIS-based product applications of the SWE reanalysis technique (Margulis et al. 2019). We also consider the impact of viewing geometry and cloud cover.

Snow CCI reanalysis is generally biased low at peak SWE (in situ snow pillows in Tuolumne, Bow and Lajoie) and relative to ASO-derived SWE estimates (Tuolumne and Aspen). However, assimilating Snow CCI fSCA improved both the



temporal evolution (Fig. 7 and 13) and spatial pattern (correlation with ASO >0.75, Sect. 3.1.2) of the reanalysis SWE
compared to in situ data (with some exceptions). The performance of Landsat reanalysis is poorer in Canadian domains than
the WUS, likely caused by the limited availability of high-quality fSCA scenes due to increased frequency of cloud cover.
Therefore, the Landsat reanalysis results only serve as a comparison reference in Canadian domains. Compared to the Landsat
results, SWE biases are linked to their associated differences in fSCA values: biases in fSCA values lead to either a longer
snowmelt period or an earlier snow-free date. We hypothesize these fSCA differences are due to differences in the products'
respective retrieval algorithms and/or the spatial resolution of the raw reflectance measurements (Sect. 3.1.3). The interannual
comparison (WYs 2001-2019) of daily basin-wide SWE shows Snow CCI posterior SWE to again be biased low compared to
that from Landsat reanalysis (all domains), with year-to-year differences in performance over the WUS tied to the number of
high-quality fSCA scenes during the ablation period (Sect. 3.1.3.3). Further studies on the fSCA quality analysis should be
conducted over the western Canadian domains.  The spatial pattern of long-term differences in peak SWE (Snow CCI vs
Landsat) indicates that biases are affected by forest cover and aspect, where Snow CCI exhibits negative differences,
particularly for densely forested regions and north-facing aspects.

The Snow CCI reanalysis presented herein aims to provide a methodology to fill the mountain SWE gap in the Snow
CCI SWE CDR. Despite its relatively coarse spatial resolution for mountain areas (0.1°), Snow CCI fSCA, which is readily
available globally, produced reasonable posterior SWE in most test domains (except for the caveats mentioned above, e.g.,
artificially early snow-free period). The main challenges when extending this framework to untested regions and/or to other
sensors (datasets) include: (1) a bias correction of fSCA since the error covariance used in the reanalysis assumes unbiased
measurements; (2) development of an algorithm to accurately identify cloud/warm surfaces from snow and setting an
appropriate cloud fraction threshold for specific regions; (3) accurate characterization of uncertainty in fSCA measurements.
Another limitation to extending this work is the limited availability of spatially and/or temporally continuous reference data.
Our temporal comparisons benefited from western North America's extensive snow pillow network, but spatial comparisons,
which relied mainly on lidar-based SWE information, were limited outside of the WUS. Ongoing work includes characterizing
the fSCA biases across different spatial and temporal (seasonal/annual) scales, correcting fSCA biases in canopy regions
(Rittger et al., 2020), and generating a dataset to fill gaps in the Snow CCI SWE CDR. Additional tasks could include a detailed
investigation of the benefits of using Snow CCI fSCA when Landsat measurements are limited (due to single Landsat tile
coverage or significant cloud cover) or combining MODIS-derived Snow CCI estimates with higher resolution estimates
derived from Sentinel-2 measurements (Bair et al., 2023; Gascoin et al., 2020).

**5 Data availability**

The Snow CCI SWE reanalysis output datasets and Landsat SWE reanalysis output datasets are publicly available at
10.5281/zenodo.13930080. The MODIS-based Snow CCI Daily SCF product (version 2) is available at http://cci.esa.int/data.
In-situ SWE measurements are available from the Natural Resources Conservation Service (NRCS) via



https://wcc.sc.egov.usda.gov/reportGenerator/ for the WUS, and for Canadian domains, access is provided through the Canadian historical Snow Water Equivalent (CanSWE) dataset (https://doi.org/10.5281/zenodo.7734616).

## 6 Author contributions

HS contributed to implementation of the reanalysis, data analysis, manuscript conceptualization, and writing, YF contributed to implementation of the reanalysis, manuscript conceptualization and revision. SAM contributed to manuscript conceptualization, revision, and supervision. CM contributed to data acquisition, manuscript conceptualization, writing, and revision. LM contributed to data acquisition, manuscript conceptualization, writing, and revision. CD contributed to data acquisition, manuscript conceptualization, writing, and revision.

## 7 Competing interests

The contact author has declared that some authors are members of the editorial board of The Cryosphere.

## 8 Acknowledgements

ECCC supported by the European Space Agency Climate Change Initiative – Snow project.

## 9 Financial support

This work is funded by the National Aeronautics and Space Administration (NASA) IDS (grant no.
80NSSC20K1293) and European Space Agency (contract no. 4000124098/18/I-NB).

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
