# Peer review of "Evaluation of the Snow CCI Snow Covered Area Product within a Mountain Snow Water Equivalent Reanalysis"

_EGUsphere, 2024_

## Author Comment (AC1)

We would like to thank the reviewers for their thorough reviews and constructive comments on the manuscript. The original comments are shown in regular black font. The responses to reviewer comments are shown in blue font, with text describing proposed additions and revisions of the manuscript shown in red font. Any original manuscript text is shown in gray font.

**Reply on Reviewer #1**

**MAJOR COMMENTS**

1. Our main concern is related to the motivation of the study. There are several snow covered area dataset available (https://lpvs.gsfc.nasa.gov/producers2.php?topic=snow). In particular NASA's MOD10 products provide similar information as the Snow CCI product and are available globally in near real time. The Snow CCI daily SCF version 2 dataset used in this study is available over the period 2000-2020 only and it seems that it is not updated (version 3 extends to 2022). We can think of some reasons but we recommend that the authors explain why they have chosen the CCI product among others.

 Response: We appreciate this comment. This study presents a use case of the Snow CCI dataset by evaluating the use of Snow CCI snow cover fraction product to fill gaps in the Snow CCI SWE dataset. The Snow CCI product is part of the European Space Agency (ESA) initiative to generate historical records of essential climate variables that meet the requirements of the Global Climate Observing System. The Snow CCI project provides both global snow cover fraction and snow water equivalent (SWE) products. However, due to known limitations, a complex terrain mask is applied to Snow CCI SWE product. The choice of Snow CCI snow cover fraction product ensures the consistency with the overall dataset development framework and aligns with the objectives of the Snow CCI project. The MOD10 snow cover fraction product and others exist with potential differences in product characteristics. While further studies could explore these products, such a comparison is beyond the scope of this study. We acknowledge other products and emphasize that this work is specifically focused on the evaluation of the Snow CCI and that other products could be explored in the Introduction and Conclusions sections respectively:

In the Introduction (L.56-58), we propose to add a clarification: A snow cover fraction product derived from optical satellite data spanning multiple decades already exists within the Snow CCI program, which provides a natural connection to a mountain SWE product via the BSRF. We acknowledge the existence of other snow cover datasets, such as MODIS-based MOD10 SCF product (Hall and Riggs, 2016), which utilize different retrieval approaches. However, this study specifically focuses on evaluating the Snow CCI SCF product within the BSRF framework to maintain consistency with the overall dataset development framework and objectives.

In the Conclusion (L.581), we propose to add an acknowledgement: "While this study focuses on the Snow CCI snow cover dataset, future work should explore comparisons with other snow cover products, to assess their applicability and optimality for SWE estimation in mountainous terrain."

2. In addition, this study shows that a Landsat-derived SWE reanalysis largely outperforms the MODIS-CCI-derived reanalysis. Therefore, we are tempted to conclude that a global mountain snow reanalysis should be performed with Landsat fSCA. But the authors seem to implicitly consider that this is not an option. We believe that this should also be clearly stated and justified in the introduction.

Response: We appreciate the reviewer's comment regarding the performance of Landsat-derived SWE reanalysis and the potential of using Landsat fSCA for a global mountain snow reanalysis. The aim in this paper is not to indicate that SnowCCI fSCA should be used instead of Landsat fSCA, but rather explore the potential (and limitations) of using a fully consistent set of Snow CCI products. Future work aimed at a global reanalysis should ideally include all available fSCA products and their relative uncertainties.

While we acknowledge the advantages of Landsat data, there are potential challenges that make its use for a global reanalysis an open question worthy of future work beyond the scope of this study. Landsat's 16-day revisit period may be insufficient to capture dynamic snow cover changes, particularly in regions with frequent melt events or at high latitudes. Furthermore, cloud cover often leads to substantial data gaps, limiting its temporal completeness. Specifically, in Canadian domains, Landsat may not always outperform other products due to its coarse temporal resolution and frequent cloud cover.

In this study, due to the limited availability of spatially distributed verification SWE datasets, we generated a Landsat-derived SWE reanalysis, which performs well in the Western United States (WUS), to serve as a comparison reference. By comparing Snow CCI snow cover with Landsat snow cover, we were able to attribute SWE differences to snow cover differences and analyze the importance of snow cover fraction in SWE reanalysis.

In the Conclusion (L.568), we propose to add: "The Snow CCI reanalysis presented herein aims to provide a methodology to fill the mountain SWE gap in the Snow CCI SWE CDR. Rather than suggesting that Snow CCI fSCA should be used instead of Landsat fSCA, this study explores the potential and limitations of using a fully consistent set of Snow CCI products."

**MINOR COMMENTS**

Several acronyms were not defined (L14 WY, L69 fSCA, L11 CCI, L249 DOWY)

Response: Revised. fSCA is now defined in L82.

Fig 2: Because the tiles are defined in lon/lat angles, Fig. 2e merges tiles of different areas, giving more weight to tiles close to the equator.

Response: Cloud fraction is calculated for each tile based on the raw Snow CCI snow cover image. It is only dependent on time for each tile. We acknowledge that when aggregating tile-wise cloud fraction data without accounting for the varying physical areas of tiles, larger-area

tiles (typically those closer to the equator) contribute more to the result. Through a rough calculation, normalizing the data by the actual physical area of each tile yields a cloud fraction threshold of approximately 0.55, which is close to 0.6.

Additionally, the WUS domains, located at lower latitudes, generally exhibit lower cloud fractions, resulting in a higher cloud fraction threshold compared to Canadian domains at higher latitudes. To ensure data quality, we apply a stricter cloud threshold for the Canadian domains to filter out more images with cloud cover.

L200 In this earlier study MODSCAG algorithm was used to retrieve fSCA and not SCAmod. Therefore there is no reason to specifically refer to this study to justify the 15% value. Other evaluations of MODIS-based snow products should be considered.

Response: The Snow CCI snow cover product includes an uncertainty layer that provides an unbiased RMSE estimate (that neglects systematic errors). For example, the average internal measurement error for the Snow CCI product is approximately 8% over the Tuolumne Basin. During the initial phase of this study, we conducted a sensitivity test on the measurement error and found that assuming a 15% measurement error for the Snow CCI (MODIS) product resulted in slightly improved performance, but without a significant sensitivity to the measurement error. The attached figure illustrates the comparison of posterior (post) peak SWE relative to ASO SWE under different measurement error assumptions.

[Figure]

Figure R1. Bar plot of peak SWE statistics for comparison with ASO SWE in the Tuolumne domain, under different measurement error assumptions.

We acknowledge that a uniform and constant 15% measurement error is a simplification that should be studied more, where ideally a space-time varying estimate of the uncertainty is built into the Snow CCI product in a way that also accounts for potential biases.

In Conclusion (L.551), we propose to add: "Since Snow CCI fSCA is retrieved based on reflectance observations from the MODIS sensor, we applied the measurement error (standard deviation of 15%) used for other MODIS-based product applications of the SWE reanalysis technique (Margulis et al., 2019). We acknowledge that a 15% measurement error at nadir viewing is a simplification and that ideally the space-time varying estimate uncertainty built into the Snow CCI product would describe the uncertainty in the SCF estimates in a way that is meaningful for our data assimilation use case."

L206: "the weighting function $w(\theta)$ varies within (0,1] by its definition" Yet maximum MODIS scan angle is 55° hence w will never reach 0. It is difficult to understand how this weighting factor w was defined by Dozier et al. 2008. It would be useful to plot w as a function of the MODIS scan angle. In addition, from a more practical perspective, how were obtained the MODIS zenith angle values? It seems that the Snow CCI product does not provide such information.

Response: Thank you for your comment. To clarify, we propose to plot $w$ as a function of the MODIS scan angle in a second y axis in Figure 3 for the revised manuscript (see below).

MODIS sensor zenith angle was obtained from the MOD09GA product, which can be downloaded from NASA Earthdata Search (https://search.earthdata.nasa.gov/search?q=MOD09GA&long=0.0703125).

On L.190, we propose to add a clarification: "… where the measurement error covariance $C_v^{Snow\ CCI}(\theta)$ is a function of the MODIS sensor viewing angle $\theta$, obtained from the MOD09GA product."

[Figure]

Figure 3. Left y axis: the impact of the $w(\theta)$ threshold on the accuracy, i.e. measurement error standard deviation, of Snow CCI fSCA for assimilation. Right y axis: the function of $w(\theta)$. Areas below the threshold of $w = 0.2$ are excluded from the assimilation. The measurement error of Landsat fSCA is represented by the red triangle (i.e., 0.1 at nadir).

L274. Cite the Vionnet et al. paper instead of the URL.

Response: Suggestion adopted.

L280. The interpolation method is first an "aggregation" and then a nearest neighbor interpolation. What means aggregation (average?). Why not resampling directly to the target grid in a single operation? Why a nearest neighbor interpolation?

Response: The ASO SWE data is available at a resolution of 50 meters, while the static topographic data (retrieved from NASA Shuttle Radar Topography Mission (SRTM) DEM) used in the reanalysis is available at a resolution of 30 meters. To harmonize these datasets, we first interpolate the ASO data onto the higher resolution 30-meter grid (SRTM grid) using the nearest neighbor method. This approach introduces minimal error, as the resolutions of the two datasets are similar. We then aggregate the ASO data to the coarser 0.01-degree resolution by averaging the subgrid SWE values.

L295. Why was the evaluation limited to peak SWE? There are many other ASO SWE products in the Tuolumne (e.g. 49 SWE products between 2012 and 2019, Sourp et al. 2024 https://doi.org/10.5194/egusphere-2024-791, data available online https://nsidc.org/data/aso_50m_swe/versions/1).

Response: We conducted evaluations on multiple days (using the additional data you cite) throughout the study period, and the performance on these days was consistent with the performance observed on the peak SWE day. The table below provides a summary of the statistics for all days when ASO SWE data was available. We propose to include this additional information in the Supplemental Information section of the revised manuscript.

The evaluation focused on peak SWE because it represents a critical metric for water resource management and hydrologic forecasting.

| ASO basin | Water Year | Day of Water Year | Correlation | | | RMSD (m) | | |
|---|---|---|---|---|---|---|---|---|
| | | | prior | Landsat posterior | Snow CCI posterior | Prior | Landsat posterior | Snow CCI posterior |
| Tuolumne | 2015 | 140 | 0.58 | 0.82 | 0.7 | 0.07 | 0.1 | 0.1 |
| | | 156 | 0.61 | 0.86 | 0.76 | 0.09 | 0.09 | 0.1 |
| | | 176 | 0.53 | 0.86 | 0.78 | 0.08 | 0.06 | 0.07 |
| | | 185 | 0.48 | 0.83 | 0.75 | 0.08 | 0.05 | 0.06 |
| | | 191 | 0.45 | 0.81 | 0.72 | 0.12 | 0.07 | 0.09 |
| | | 197 | 0.45 | 0.82 | 0.73 | 0.11 | 0.06 | 0.08 |
| | | 209 | 0.44 | 0.79 | 0.7 | 0.11 | 0.07 | 0.09 |
| | | 213 | 0.47 | 0.79 | 0.7 | 0.07 | 0.04 | 0.05 |
| | | 240 | 0.62 | 0.81 | 0.66 | 0.07 | 0.05 | 0.06 |
| | | 251 | 0.7 | 0.72 | 0.52 | 0.03 | 0.03 | 0.03 |
| | 2016 | 178 | 0.66 | 0.9 | 0.84 | 0.41 | 0.26 | 0.37 |
| | | 183 | 0.64 | 0.89 | 0.85 | 0.37 | 0.22 | 0.32 |
| | | 190 | 0.62 | 0.9 | 0.84 | 0.36 | 0.21 | 0.32 |
| | | 199 | 0.65 | 0.91 | 0.85 | 0.34 | 0.18 | 0.28 |
| | | 209 | 0.68 | 0.92 | 0.83 | 0.39 | 0.24 | 0.34 |
| | | 222 | 0.69 | 0.93 | 0.86 | 0.36 | 0.19 | 0.29 |
| | | 240 | 0.56 | 0.89 | 0.87 | 0.28 | 0.15 | 0.19 |
| | | 282 | 0.77 | 0.82 | 0.6 | 0.04 | 0.03 | 0.04 |
| | 2017 | 154 | 0.59 | 0.88 | 0.8 | 0.45 | 0.3 | 0.48 |
| | | 183 | 0.54 | 0.92 | 0.87 | 0.6 | 0.27 | 0.46 |
| | | 214 | 0.6 | 0.91 | 0.85 | 0.58 | 0.29 | 0.5 |
| | | 282 | 0.31 | 0.82 | 0.72 | 0.28 | 0.17 | 0.2 |
| | 2018 | 205 | 0.59 | 0.86 | 0.79 | 0.3 | 0.19 | 0.4 |
| | | 240 | 0.51 | 0.82 | 0.73 | 0.17 | 0.11 | 0.35 |

L309-311. We find a bit confusing to use the Landsat posterior SWE as reference in section 3.1 especially in Figure 6 (where the colors indicate the residuals with respect to Landsat reanalysis). We could suggest to replace the right panel with another scatterplot showing the prior SWE instead of the Snow CCI posterior as a y-axis.

Response: Thank you for the suggestion. The bar plot in the right panel summarizes the statistics of points represented by different colors in the left panel. It highlights the pattern that when

Snow CCI SWE aligns with Landsat SWE at the basin scale, it demonstrates improved performance relative to in-situ SWE. In other words, the largest posterior Snow CCI SWE tend to be when it departs from the Landsat posterior estimates. For comparison, the performance relative to the prior SWE is illustrated in Figure 7.

However, we find Figure 7 very informative and well designed. "30 out of 59 sites-year show improvement relative to prior" Does it suggest that assimilating the Snow CCI product was not beneficial on average?

Response: Thank you for catching this. While 30 out of 59 (in situ) site-years show improvement relative to the prior, it does not necessarily imply that assimilating the Snow CCI product was not beneficial on average. The results depend on factors such as in-situ site characteristics, variability in snow conditions, or specific site-year combinations. For example, assimilating Snow CCI product improves the correlation across all snow pillow sites in the Tuolumne in 2007, whereas the improvement in 2003 is less significant. Furthermore, it is important to note that available in-situ SWE measurements are relatively sparse and may not always be representative for comparison with gridded SWE estimates at a 0.01-degree resolution.

For this reason, we would like to clarify that the ASO comparison (Figure 8 and Table 3) provides a more representative assessment since it is a spatially-distributed product. This comparison demonstrates that assimilating Snow CCI fSCA improves the correlation relative to the prior, highlighting the overall benefits of the Snow CCI product.

Fig. 9: 1%-99% percentiles are usually taken to represent large sample size, here there are only 20 values.

Response: Thank you for your comment. We acknowledge that this range is typically applied to large sample sizes. To address your concern, we explored alternative ways to represent variability, such as using the full range (minimum – maximum) to ensure clarity for the given sample size. We propose to revise Figure 9 to:

[Figure]

Similarly, we propose to revise Figure 14 to:

[Figure]

L470. Figure 11 suggests that the thresholds of cloudiness and *w* discussed earlier in the paper could be revisited. This could be discussed and ideally a sensitivity analysis to these thresholds would be useful (but it may be a lot of computation to ask).

Response: Thank you for your suggestion. We agree that revisiting the thresholds for cloudiness and sensor viewing angle (through *w*) could potentially improve the number of good-quality snow cover fraction image. While a full sensitivity analysis would indeed be valuable, it is beyond the scope of this study due to the computational resources required. We propose to revise the discussion of Figure 11 (L.467) and conclusion (L.551) to: "Future work could explore sensitivity tests of the relative differences regarding the thresholds of cloudiness and sensor viewing angle discussed in Sect. 2.3."

L526. Fig14 How to interpret the poor performance (i.e. the large difference with Landsat posterior estimates) in Bow domain for forest cover 0-10% in comparison with 10-50%?

Response: Thank you for pointing this out. Upon further examination, the poor performance in the Bow domain for forest cover 0-10% compared to 10-50% appears to be attributable to outlier data points. To address this, we have added a new histogram plot showing the distribution of

relative differences across all forest cover bins. For the 0-10% forest cover range, the mean value is notably smaller than the median, driven by a few pixels with significant negative relative differences. These pixels are rare and represent outliers in the distribution. When considering the median value, the performance for the 0-10% forest cover range is comparable to that of the 10-30% range.

In the revised manuscript, we propose to clarify this on L.526:

"…, Bow forest cover 0-10% excepted. Note that the bar plots show mean relative differences in each bin as functions of forest cover and aspect. The poor performance in the Bow domain for forest cover 0-10% is likely due to outlier data points. However, when considering the median relative differences in each bin, the performance for the 0-10% forest cover is comparable to that of the 10-30% range."

[Figure]

L568. 0.01°

Response: Suggestion adopted.

---

## Author Comment (AC2)

We would like to thank the reviewers for their thorough reviews and constructive comments on the manuscript. The original comments are shown in regular black font. The responses to reviewer comments are shown in blue font, with text describing proposed additions and revisions of the manuscript shown in red font. Any original manuscript text is shown in gray font.

**Reply on Reviewer #2**

**MAJOR COMMENTS**

1. My biggest critique of the paper is that it does not appear to have a significant or well-understood result. For those who are interested in the CCI snow dataset, the performance of the snow reanalysis are mixed, showing less accurate results than the Landsat-based snow dataset. The reasons are not fully explored here but several ideas are speculated (see Conclusions section, L. 557-566). I think it can be useful to have papers that show "negative results", but it seems the paper stops short in showing why those poorer results are achieved. Exploring one or more of those hypotheses would add more substance and could yield a broader result that goes beyond the nuances of the CCI dataset. For instance, addressing the question of retrieval algorithm vs. spatial resolution (L. 560) would benefit a wider audience (e.g., those who are interested in CCI data as well as those interested in other remotely sensed snow cover datasets). Alternatively, testing the impact of the weighting scheme (versus using no weighting) on the SWE reanalysis might be another contribution that could be made here within the scope of the analysis, and this could have broader appeal.

Response: Thank you for your constructive critique. We appreciate your feedback regarding the significance of the results. While we agree that exploring the reasons behind the poorer results or testing hypotheses (e.g., retrieval algorithm vs. spatial resolution or the impact of the weighting scheme) would be valuable, these topics are beyond the scope of this paper, which is centered specifically on the evaluation the Snow CCI product. Our goal is to assess the applicability of Snow CCI within an existing snow reanalysis framework, providing insights that can inform future research and product improvements.

In this study, we are an "application user" of an existing product (in this case, for snow reanalysis) rather than algorithm developers. We acknowledge the limitations and highlight potential areas for future work, such as diving deeper into the retrieval algorithm and spatial resolution issues. By maintaining our focus on the evaluation of the Snow CCI product in the context of an "application user", we aim to provide a foundation for further research that can explore broader questions in greater detail.

Future work should first compare fSCA products themselves (independent) of using them as an input to a SWE reanalysis. Since we are users rather than developers of the Snow CCI product, our role is to test its application in SWE reanalysis rather than to conduct an in-depth investigation of its underlying methodology. While such comparisons have been done within

SnowPex (The Satellite Snow Product Intercomparison and Evaluation Exercise) and Snow CCI reports, they have unfortunately not yet made it into peer-reviewed publications. We suggest that the Snow CCI team or other fSCA product developers further examine these important issues in future research based on the conclusions of this work.

To address this major comment, we propose to revise the Introduction to make objectives more explicit.

**Introduction (L.52-65):**

"This study is motivated by the need to fill the mountain SWE gap in the Snow CCI SWE product. Specifically, we explore the use of a Bayesian snow reanalysis framework (BSRF) previously implemented in various mountain regions across the globe including the Sierra Nevadas, the Andes, and High Mountain Asia (Margulis et al., 2016, 2019; Cortés et al., 2016; Liu et al., 2021; Fang et al., 2022). The framework combines prior snow estimates from an ensemble of land surface model simulations with satellite derived fractional snow-covered area (fSCA) to generate retrospective time series of snow extent and SWE (Margulis et al. 2019). The previous implementations have used the Landsat-based SCF product (Painter et al., 2003; Cortés et al., 2014) and the MODIS-based MODSCAG SCF product (Painter et al., 2009; Margulis et al., 2019). Because a snow cover fraction product (MODIS-based) also exists within the Snow CCI program, it is a natural choice to use to develop a mountain SWE product within the same program. We acknowledge the existence of other snow cover datasets, such as MODIS-based MOD10 SCF product (Hall and Riggs, 2016), which utilize different retrieval approaches. However, this study specifically focuses on evaluating the Snow CCI SCF product within the BSRF framework to maintain consistency with the overall dataset development framework and objectives. The fSCA products specified above can differ in their input data source, resolution, and retrieval approach, and as a result, they yield different fSCA values and will have different error characteristics.

~~The primary objective of this study is to evaluate the use of Snow CCI SCF products in the BSRF for estimating SWE in mountainous terrain at Snow CCI SCF grid resolution of 0.01 degrees (~1km), thereby potentially filling a key gap in the existing Snow CCI SWE product.~~ Therefore, the primary objective of this study is to evaluate whether using the Snow CCI SCF product within the BSRF can provide meaningful SWE estimates in mountain terrain. The Snow CCI SCF products are global in coverage, so this approach can potentially be extended to all mountain regions in the future. We evaluate this objective by implementing the product across four test watersheds. Two watersheds are in the WUS where the BSRF was already previously implemented using Landsat (Fang et al., 2022). For these regions we can compare the established performance of the Landsat implementation to performance when using the SnowCCI fSCA in order to characterize how differences in the fSCA products result in different SWE estimates. We also implement these same choices of fSCA data in two new watersheds in Canada. While we will also compare the performance of the two fSCA datasets within these new watersheds, by extending the BSRF to a new region of the globe we have an opportunity to assess challenges that would arise in a global implementation of the BSRF. The remainder of this paper is

organized as follows. Section 2 describes the methods, data, and application domain, Section 3 provides results and discussion, and Section 4 provides the key conclusions of the study."

**Conclusions (first two paragraphs L.545-566):**

This study explored the potential for using the Snow CCI fSCA Climate Data Record (CDR) for mountain SWE estimation using a Bayesian SWE reanalysis framework. Four application domains spanning the WUS and western Canada with various physiographic and climatological conditions are used for this evaluation. Since Snow CCI fSCA is retrieved based on reflectance observations from the MODIS sensor, we applied the measurement error (standard deviation of 15%) used for other MODIS-based product applications of the SWE reanalysis technique (Margulis et al. 2019). We acknowledge that a simple 15% measurement error is a simplification and that ideally the space-time varying estimate uncertainty built into the Snow CCI product would describe the uncertainty in the SCF estimates in a way that is meaningful for our data assimilation use case. We also consider the impact of viewing geometry and cloud cover.

The Snow CCI reanalysis is generally biased low at peak SWE (in situ snow pillows in Tuolumne, Bow and Lajoie) and relative to ASO-derived SWE estimates (Tuolumne and Aspen). However, assimilating Snow CCI fSCA improved both the temporal evolution (Fig. 7 and 13) and spatial pattern (correlation with ASO >0.75, Sect. 3.1.2) of the reanalysis SWE compared to in situ data (with some exceptions). The performance of Landsat reanalysis is poorer in Canadian domains than the WUS, likely caused by the limited availability of high-quality fSCA scenes due to increased frequency of cloud cover. Therefore, the Landsat reanalysis results only serve as a comparison reference in Canadian domains. Compared to the Landsat results, SWE biases are linked to their associated differences in fSCA values: biases in fSCA values lead to either a longer snowmelt period or an earlier snow-free date. We hypothesize these fSCA differences are due to differences in the products' respective retrieval algorithms and/or the spatial resolution of the raw reflectance measurements (Sect. 3.1.3). The interannual comparison (WYs 2001-2019) of daily basin-wide SWE shows Snow CCI posterior SWE to again be biased low compared to that from Landsat reanalysis (all domains), with year-to-year differences in performance over the WUS tied to the number of high-quality fSCA scenes during the ablation period (Sect. 3.1.3.3). Further studies on the fSCA quality analysis should be conducted over the western Canadian domains. The spatial pattern of long-term differences in peak SWE (Snow CCI vs Landsat) indicates that biases are affected by forest cover and aspect, where Snow CCI exhibits negative differences, particularly for densely forested regions and north-facing aspects.

2.  I have concerns about interpretations if of differences in R correlation, particularly in Figures 7. A positive difference does not definitively mean that the correlation has improved, because R ranges from -1 to +1. For instance, a difference of +0.5 is not meaningful if the two R values are 0.0 and -0.5, as this indicates going from a weak negative relationship to no relationship at all. Hence, a positive difference in R is only a valid indicator of improved correlations when both are greater than 0. Why not just use R^2 here to avoid any of this potential ambiguity?

Response: Thank you for your comment. We've plotted the temporal correlation using R², and the results remain consistent with the original analysis. We propose to replace the original figures with those below based on your suggestion.

[Figure]

**Figure 7.** Temporal correlation square of Snow CCI (upper left) and Landsat (upper right) posterior daily SWE vs. snow pillow daily SWE measurements at snow pillow sites in the Tuolumne domain. The bottom panels display differences between the posterior correlation and the prior correlation (posterior $R^2$ – prior $R^2$). Station-years with improvements are in the black boxes. Cases where snow pillow measurements are incomplete and/or annual peak SWE values were lower than 2 cm are greyed out.

[Figure]

**Figure 13.** Same as Fig. 7, but for Bow and Lajoie domains with snow pillows names in black and blue, respectively.

**GENERAL COMMENTS**

The manuscript uses two different acronyms to describe the same variable – namely, "fSCA" and "SCF" both refer to "fractional snow cover". This could cause confusion, so I suggest picking one convention and using it exclusively.

Response: Thank you for your suggestion. We agree this could cause confusion. We use SCF only to refer to the original Snow CCI snow cover fraction (SCF) product name (which uses "SCF"). On line 81-82 in the original text, we clarified that fSCA is used elsewhere "The Snow CCI SCFV dataset will be referred to as Snow CCI fSCA hereafter.".

The paper could use additional description on the land surface model and PBS approach. The description (L. 129-132) is rather meager. This expanded description does not have to be highly detailed but should provide enough context and explanation to help any readers who are unfamiliar with the approaches applied in Fang et al. (2022), etc. Additionally, the paper should confirm/clarify whether there are any other differences in the methods other than using a different snow cover dataset (see L. 283-286).

Response: Thank you for your comment. There are no differences in the application beyond the use of a different fSCA dataset (and the necessary cloud/viewing angle thresholds), which is why we primarily referred to previous work for description of the methods. We propose to add more description on the LSM and PBS approach to the original text on L.129-132 from:

"In this study, the land surface model inputs and uncertainty parameters for prior ensemble perturbations are following Fang et al. (2022), which were derived specifically for the WUS. Historical remotely sensed fSCA measurements from Snow CCI are assimilated to update the prior estimates via a Particle Batch Smoother (PBS) scheme to yield SWE reanalysis estimates."

to:

"The snow reanalysis framework used herein is the same as in previous applications, but with use of the Snow CCI fSCA data. The method combines a spatially distributed Land Surface Model (LSM) and a Particle Batch Smoother (PBS) to estimate snow dynamics. We applied the LSM, specifically the Simplified Simple Biosphere -Snow Atmosphere Soil Transfer (SSiB-SAST) model, to simulate SWE, snow density, and snow depth. The Liston Snow Depletion Curve (SDC) model is coupled with the LSM to predict fSCA based on modeled SWE and its sub-grid heterogeneity. The LSM-SDC accounts for prior uncertainties from meteorological forcing, model parameters, and sub-grid snow variability. Uncertainty models and parameters are described in Fang et al. (2022). In the prior step, these uncertainties are embedded in an ensemble of model estimates. A Particle Batch Smoother (PBS) method assimilates fSCA measurements. This involves assigning likelihood-based weights to ensemble members, producing posterior snow estimates with improved accuracy."

Figures 4 and 5 (and corresponding text) seem to be out of order. I think the authors should consider swapping the order for both figures and their text. If I understand the process correctly, you would first do the screening (Fig. 5) and then do the assimilation (Fig. 4).

Response: Thank you for the suggestion. We will swap the order in the revised manuscript based on this suggestion.

More description is needed on whether and how quality control was done for the snow pillow data (Section 2.4.1). The California snow pillows are infamously noisy compared to NRCS SNOTEL (and note there are zero NRCS SNOTEL sites in this study). One of the sites used (Dana Meadows) also has known data issues during the study period (e.g., a tree growing in the snow pillow in 2007, see Lundquist et al., 2015).

Response: Thank you for pointing out the need for additional clarification regarding the quality control of in-situ SWE data. We acknowledge that snow pillows in California can be noisier than NRCS SNOTEL data, as noted in prior studies (e.g., Lundquist et al., 2015). The in-situ SWE data used in this study was downloaded from the NRCS portal (See L.274: In-situ SWE measurements are available from the Natural Resources Conservation Service (NRCS) via https://wcc.sc.egov.usda.gov/reportGenerator/). The data published on this portal combines SNOTEL data from NRCS and snow pillows from CDEC. In this study, we implemented a data screening approach consistent with the method outlined in Fang et al. (2022). This approach includes removing in-situ SWE measurements that are incomplete and/or annual peak SWE values lower than 2 cm.

Additionally, while snow pillow data were used as an independent verification source, we placed greater emphasis on the comparison with ASO data, which provide high-resolution and accurate gridded SWE estimates. The ASO data is critical for validating the performance of the reanalysis framework results and addressing potential limitations in point-scale snow pillow data.

In a few places in the "Results and Discussion" section, the possible impacts of wildfire are speculated to be a factor (L.362-364, 439-441) but without compelling evidence. In general, I think the discussion and interpretation of results could be improved through more direct/substantive connections to other relevant studies.

To address this concern, we propose to remove the referenced speculated factors.

We propose to revise the L.362-364:

"This may be due to the insufficient characterization of snow albedo uncertainty in Colorado (Fang et al., 2022), where studies have shown that snow albedo is influenced by factors such as dust, black carbon, and other light-absorbing particles (Deems et al., 2013)."

Deems, J. S., Painter, T. H., Barsugli, J. J., Belnap, J. & Udall, B. Combined impacts of current and future dust deposition and regional warming on Colorado River Basin snow dynamics and hydrology. Hydrol. Earth Syst. Sci. 17, 4401–4413 (2013).

We propose to revise the L.439-441 to:

"In the Aspen domain, peak SWE generally occurs in April, with February and March receiving substantial snowfall. While the seasonal cycle of SWE is comparable to the Landsat reference (Fig. 9a), WY 2018 is a case of poor performance, but with significant positive biases in posteriori SWE. Correspondingly, Snow CCI predicts an abnormally long snowmelt season (April to September) characterized by significant positive biases in fSCA compared to the Landsat reference, particularly before July (Fig. 10b).  This is a unique case where Snow CCI fSCA exhibits low quality (biased) estimates in the Aspen domain, and the specified measurement error cannot correct biases in fSCA. Future efforts at deeper investigation into the retrieval algorithm are warranted."

**LINE COMMENTS**

L. 52-58: Here, I think a sentence is clarify that this is not the same dataset as the UCLA Western U.S. Reanalysis daily snow dataset (published at NSIDC). The methods appears to be the same, but the source snow cover dataset is changed here to support the CCI effort. I did not fully understand that these were different datasets until I got deeper into section 2. I suspect others who are familiar with the UCLA WUS dataset may also experience some confusion.

Response: Thank you for the suggestion. We propose to revise the text to: "This study is motivated by the need to fill the mountain SWE gap in the Snow CCI SWE product. Specifically, we explore the use of a previously implemented Bayesian snow reanalysis framework (BSRF; (Margulis et al., 2016, 2019; Cortés et al., 2016; Liu et al., 2021; Fang et al., 2022). The framework combines prior snow estimates from an ensemble of land surface model simulations with satellite-derived fractional snow-covered area to generate retrospective time series of snow extent and SWE (Margulis et al., 2019). It is important to note that while the BSRF methodology is consistent with that used in the UCLA Western U.S. Reanalysis daily snow dataset (published at NSIDC), the datasets themselves are distinct."

L. 93: You could note here that not only are Lajoie and Bow River basins at different latitudes, but also different snow climates.

Response: Thank you for your suggestion. We propose to revise the text to: "The Lajoie and Bow River basins are higher-latitude forested basins that have not been explored in previous applications of the BSRF. Additionally, these basins represent different snow climates, offering an opportunity to evaluate the framework's performance across varying snow regimes."

L. 98: Remove "The" before "Aspen".

Response: Suggestion adopted.

L. 109-113: This is somewhat redundant with what is described earlier at L. 59-63. Why repeat this information about the goal? Consider some merging/reorganization of this text with the earlier text.

Response: Thank you for catching that. We acknowledge that repeating the information about the goal is unnecessary. We propose to revise L.109-113 by deleting the repetitive information: ""

L. 127: Why "mostly"? What else is used other than Landsat?

Response: We propose to remove "mostly" in the revised version.

L. 133: I am not sure "scatters" is the right word. Please consider rephrasing.

Response: We propose to rephrase L.133: "The preprocessing of Snow CCI fSCA for SWE reanalysis, including cloud screening (Sect. 2.3.1) and viewing geometry screening (Sect. 2.3.2), is carefully conducted to avoid misclassifying clouds, forests, and other non-snow features as snow."

L. 144: Replace "at" with "in".

Response: Suggestion adopted.

L. 157: Please add a little more description about how this cloud mask is produced in snow CCI.

Response: Thank you for your suggestion. We propose to add this sentence to L.157: "The cloud mask in the Snow CCI product is derived using an adapted version of the Simple Cloud

Detection Algorithm 2.0 (SCDA2.0) (Metsämäki et al., 2015). This algorithm is based on the brightness-temperature difference between 11 µm and 3.7 µm, with clouds exhibiting significantly large negative values."

L. 184, 199: The Rittger et al. (2020) citation is relevant and could be included at this parts of the text.

Response: We will cite this paper in the revised version: Rittger, K., Raleigh, M. S., Dozier, J., Hill, A. F., Lutz, J. A., and Painter, T. H. (2020). Canopy adjustment and improved cloud detection for remotely sensed snow cover mapping. Water Resour. Res. 56, e2019WR024914. doi:10.1029/2019WR024914

L. 210 Consider replacing "reliability" with "quality".

Response: Suggestion adopted.

L.214-220 and Figure 3: What is theta angle for w(theta) ~= 0.2? It might help to add a second (non-linear) x-axis with the theta values, as this might be easier for some to interpret (i.e., satellite view angle).

Response: Thank you for your suggestion. We propose to add a second y-axis with the theta values. The theta angle for w(theta)~=0.2 is around 50 degrees.

[Figure]

Figure 3. Left y axis: the impact of the w(θ) threshold on the accuracy, i.e. measurement error standard deviation, of Snow CCI fSCA for assimilation. Right y axis: the function of w(θ). Areas below the threshold of w = 0.2 are excluded from the assimilation. The measurement error of Landsat fSCA is represented by the red triangle (i.e., 0.1 at nadir).

L. 216: Please provide more information on where/when these calculations were performed.

Response: As stated in L.215, the measurement error is calculated using Eq. 3. The data assimilation framework requires the specification of the fSCA error standard deviation as an input. As outlined in Eq. 13 and Eq. 14 of Margulis et al. (2015), the fSCA error is used when calculating the posterior weight of the particles.

L. 245: Again, it is hard for a reader to know what theta angle corresponds to weights of 0.2 or less. Describing in terms of theta is more straightforward, in my opinion.

Response: We propose adjusting Figure 3 (see above) to address this concern.

L. 262: Consider adding some citations here on spatial representativeness, such as Meromy et al. (2012) or Herbert et al. (2024).

Response: Thank you for your suggestion. We will cite these papers in the revised version.

Meromy, L., Molotch, N. P., Link, T. E., Fassnacht, S. R., & Rice, R. (2012). Subgrid variability of snow water equivalent at operational snow stations in the western USA. Hydrological Processes. https://doi.org/10.1002/hyp.9355

Herbert, J. N., Raleigh, M. S., & Small, E. E. (2024). Reanalyzing the spatial representativeness of snow depth at automated monitoring stations using airborne lidar data. The Cryosphere, 18(8), 3495–3512. https://doi.org/10.5194/tc-18-3495-2024

L. 275: This would read better if rephrased as "in the Aspen domain".

Response: Suggestion adopted.

L. 277-278: How did you handle the aggregation at the basin boundaries of the ASO data? In other words, some 0.01 deg pixels did not have complete ASO coverage and likely have missing data at the finer scale. Please describe in more detail your methods and assumptions here.

Response: We set pixels without complete ASO coverage or with missing data at the finer scale to NaN to ensure that the ASO data does not introduce artifacts along the basin boundary.

L. 284: Add "(~1 km)" after "0.01 deg".

Response: Suggestion adopted.

L. 287: You could add that the reason why it is not well understood over Canada is because it has not been produced there before (e.g., UCLA Western US reanalysis does not include Canada).

Response: As suggested, we propose to revise L.287 to: "The performance of the Landsat reanalysis SWE is well-understood over the western US but not over western Canada, as it has not been produced there previously."

L. 300: Add "more" before "forested".

Response: Suggestion adopted.

L. 301: Add "(Fig. 1)" after "WUS".

Response: Suggestion adopted.

L. 307: Specify the type of correlation (e.g., pearson, spearman, …)

Response: We calculate the Pearson correlation coefficient. We will clarify it in the title of Figure 6: Bar plots show variations in Pearson correlation and RMSD

L. 318: I think this reads better if rephrased to say "… peak SWE better matches the Landsat reference …"

Response: Suggestion adopted.

L. 332: Suggest replacing "sites" with "pixels" because you are talking about the gridded dataset.

Response: Suggestion adopted.

L. 357: Suggest using more precise language here: replace "less snowy" and "more snowy" with "lower SWE" and "higher SWE".

Response: Suggestion adopted.

L. 360: Replace "in Colorado" with "domain".

Response: Suggestion adopted.

L. 404: Remove "ASO" as it is not relevant here.

Response: Suggestion adopted.

L. 435: Double check. Should this be "magenta" rather than "blue"?

Response: Thank you for your comment. We double checked that two sample WYs with comparable performance are highlighted with blue boxes (i.e., 2002 and 2010 for the Tuolumne, 2006 and 2012 for the Aspen) while two typical WYs of differing performance are highlighted with magenta boxes (i.e., 2006 and 2011 for the Tuolumne, 2011 and 2018 for the Aspen).

L. 529-531: Another complicating factor here with the thermal temperature screening (L. 393-396) is the mixed pixel problem with temperatures from multiple sources (snow, trees, etc.). Consider including this in your discussion and see Lundquist et al. (2018).

Response: Thank you for your suggestion. We propose to revise L.529-531: "Another hypothesis, as discussed in 3.1.3.2, is that small snow-covered areas (low fSCA) may still exist on north-facing slopes during the warm season that are captured by Landsat but are set to zero in Snow CCI by the temperature threshold screening (Metsämäki 2015). Additionally, forest-covered areas have a mixed-pixel temperature problem. SCAmod applies a snow temperature threshold to the combined components in a grid cell, but the temperatures of individual end-members (e.g., snow) provide more valuable information. Lundquist et al. (2018) developed a method to separate snow and forest temperatures using multispectral un-mixing, leveraging differences in midwave and longwave infrared bands. Future studies could explore and apply this method across broader areas to address the mixed-pixel temperature issue."

L. 551: Consider adding ", but not canopy correction (Rittger et al., 2020)." at the end of this sentence.

Response: Suggestion adopted.

L. 555-556: Another factor that could be discussed is higher forest cover in the Canadian domains. This should not be neglected.

Response: Thank you for your suggestion. We propose to add higher forest cover at the end of the sentence: "The performance of Landsat reanalysis is poorer in Canadian domains than the WUS, likely caused by the limited availability of high-quality fSCA scenes due to increased frequency of cloud cover and higher forest cover."

L. 567-571: I think this text may be too optimistic considering the results in Canada were not very skillful.

Response: We appreciate this comment. For Canadian domains, a key limitation is the limited availability of spatially and/or temporally continuous reference data. Our temporal comparisons benefited from western North America's extensive snow pillow network, but spatial comparisons, which relied mainly on lidar-based SWE information, were limited outside of the WUS.

**FIGURES AND TABLES COMMENTS**

Figure 2: Why use the full water year for these distributions? I am not sure why summer (i.e., snow-free months) are relevant, especially given that Aspen may have cloudier summers than the other locations due to regular convective thunderstorms that occur over the Rockies. How would this figure (and thresholds) change if ***October-June*** was used instead of the full WY?

Response: Thank you for your comments. The attached figure shows the cloud threshold if October-June is used instead of full WY. The cloud threshold is about 0.66, which is similar (slightly higher) than the previously used threshold (0.6). Some implications of using different cloud thresholds are: using higher cloud thresholds result in eliminating more images, although certain pixels remain valuable. Otherwise, using lower cloud thresholds could result in including more cloudy images that likely misclassify snow as cloud.

[Figure]

Figure 2e: I am not sure why this CDF has a step function appearance, and the text does not adequately explain it.

Response: The CDF has a step function appearance since it is based on four median values (i.e., median values of domain-wise CDF), which result in discrete jumps rather than a smooth curve.

Figure 4: What is "informative" here? This is described in the main text, but a brief description in the caption could be convenient for readers.

Response: We propose to add a brief description in the caption: Informative fSCA measurements are those that contribute to the posterior update, which only occurs when the prior ensemble spread of fSCA is greater than zero.

Figure 6 caption: Add "in" after "(circles)". Also state the units for "relative differences".

Response: Suggestion adopted.

Figure 8: Are these biases consistent with those from other ASO surveys?

Response: We conducted evaluations on multiple days throughout the study period, and the performance on these days was consistent with the performance observed on the peak SWE day. The table below provides a summary of the statistics for all days when ASO SWE data was available. The performance is consistent with those from other ASO surveys.

| ASO basin | Water Year | Day of Water Year | Correlation | | | RMSD (m) | | |
|---|---|---|---|---|---|---|---|---|
| | | | prior | Landsat posterior | Snow CCI posterior | Prior | Landsat posterior | Snow CCI posterior |
| Tuolumne | 2015 | 140 | 0.58 | 0.82 | 0.7 | 0.07 | 0.1 | 0.1 |
| | | 156 | 0.61 | 0.86 | 0.76 | 0.09 | 0.09 | 0.1 |
| | | 176 | 0.53 | 0.86 | 0.78 | 0.08 | 0.06 | 0.07 |
| | | 185 | 0.48 | 0.83 | 0.75 | 0.08 | 0.05 | 0.06 |
| | | 191 | 0.45 | 0.81 | 0.72 | 0.12 | 0.07 | 0.09 |
| | | 197 | 0.45 | 0.82 | 0.73 | 0.11 | 0.06 | 0.08 |
| | | 209 | 0.44 | 0.79 | 0.7 | 0.11 | 0.07 | 0.09 |
| | | 213 | 0.47 | 0.79 | 0.7 | 0.07 | 0.04 | 0.05 |
| | | 240 | 0.62 | 0.81 | 0.66 | 0.07 | 0.05 | 0.06 |
| | | 251 | 0.7 | 0.72 | 0.52 | 0.03 | 0.03 | 0.03 |
| | 2016 | 178 | 0.66 | 0.9 | 0.84 | 0.41 | 0.26 | 0.37 |
| | | 183 | 0.64 | 0.89 | 0.85 | 0.37 | 0.22 | 0.32 |
| | | 190 | 0.62 | 0.9 | 0.84 | 0.36 | 0.21 | 0.32 |
| | | 199 | 0.65 | 0.91 | 0.85 | 0.34 | 0.18 | 0.28 |
| | | 209 | 0.68 | 0.92 | 0.83 | 0.39 | 0.24 | 0.34 |
| | | 222 | 0.69 | 0.93 | 0.86 | 0.36 | 0.19 | 0.29 |
| | | 240 | 0.56 | 0.89 | 0.87 | 0.28 | 0.15 | 0.19 |
| | | 282 | 0.77 | 0.82 | 0.6 | 0.04 | 0.03 | 0.04 |
| | 2017 | 154 | 0.59 | 0.88 | 0.8 | 0.45 | 0.3 | 0.48 |
| | | 183 | 0.54 | 0.92 | 0.87 | 0.6 | 0.27 | 0.46 |
| | | 214 | 0.6 | 0.91 | 0.85 | 0.58 | 0.29 | 0.5 |
| | | 282 | 0.31 | 0.82 | 0.72 | 0.28 | 0.17 | 0.2 |
| | 2018 | 205 | 0.59 | 0.86 | 0.79 | 0.3 | 0.19 | 0.4 |
| | | 240 | 0.51 | 0.82 | 0.73 | 0.17 | 0.11 | 0.35 |

Figures 9 and 14: I have trouble interpreting the stacked shading here. Is there a better way to convey the ranges?

Response: We explored alternative ways to represent variability, such as using the full range (minimum – maximum) instead of 1-99$^{th}$ percentiles.

We propose to revise Figure 9 to:

[Figure]

Similarly, we propose to revise Figure 14 to:

[Figure]

Figure 10: Looking at this figure and caption alone, it is impossible to discern what the blue and magenta shading represents. This comes later in Figure 11. Need to include the information in both places so your readers can understand.

Response: We propose to add a description in the caption: Cases of consistent performance vs. inconsistent performance are highlighted in blue and magenta, respectively.

Figure 10c: Suggest stacking these two panels vertically rather than horizontally so their common axis (Difference in peak SWE) is shared/aligned.

Response: Suggestion adopted. We propose to revise Figure 10 to:

[Figure]

Similarly, we propose to revise Figure 15 to:

[Figure]

Table 1: It is incorrect to say that the Tuolumne snow pillow data come from the NRCS. These data are managed by the California Department of Water Resources via CDEC.

Response: Thank you for your comment. The in-situ SWE data used in this study was downloaded from the NRCS portal (See Line 274: In-situ SWE measurements are available from the Natural Resources Conservation Service (NRCS) via https://wcc.sc.egov.usda.gov/reportGenerator/). The data published on this portal combines SNOTEL data from NRCS and snow pillows from CDEC. We propose to revise NRCS to NRCS/CDEC in Table 1.

Table 1: It is odd to list "NRCS SNOTEL" as the source on the Aspen lines when there are zero SNOTEL sites in that domain. Suggest removing "NRCS SNOTEL" and just adding "--" in this row.

Response: Suggestion adopted.

Table 3: Suggest adding "Bias" column for both Landsat and CCI at the end.

Response: Thank you for your suggestion. We've added "Bias" column in Table 3.

| ASO basin | Year | DOWY | Correlation | | | RMSD [m] | | | Bias [m] | | |
|---|---|---|---|---|---|---|---|---|---|---|---|
| | | | Prior | Landsat Posterior | Snow CCI Posterior | Prior | Landsat Posterior | Snow CCI Posterior | Prior | Landsat Posterior | Snow CCI Posterior |
| Tuolumne | 2015 | 185 | 0.48 | 0.83 | 0.75 | 0.07 | 0.047 | 0.055 | -0.04 | -0.01 | -0.02 |
| | 2016 | 183 | 0.64 | 0.89 | 0.85 | 0.37 | 0.22 | 0.32 | -0.24 | -0.14 | -0.26 |
| | 2017 | 183 | 0.54 | 0.92 | 0.87 | 0.6 | 0.27 | 0.46 | -0.14 | -0.05 | -0.31 |
| Aspen | 2019 | 189 | 0.51 | 0.53 | 0.54 | 0.45 | 0.21 | 0.33 | 0.40 | -0.06 | -0.27 |

We think that RMSD provide a more comprehensive measure of the total deviation from ASO values. Bias, on the other hand, only captures the average difference and may cancel out positive and negative errors.

---

## Editor Decision (ED1)

2025-02-17
Submission EGUSPHERE-2024-3213

**Evaluation of the Snow CCI Snow Covered Area Product within a Mountain Snow Water Equivalent Reanalysis**

Haorui Sun et al.

One of the main comments from the reviewers focused on the motivation and how the their snow CCI product complements existing similar approaches. This was addressed by the authors working within the BSRF framework. It was also mentioned that the results were not well described, or at least to the full potential reaching out to a wider audience as suggested by one of the reviewers. The authors addressed this comment in detail, adding significant amount of results interpretation in both the introduction and conclusion for a more explicit understanding of the paper's objectives. Finally there was a comment about the interpretation of R2 values and their interpretation, where a figure was modified in the revised version facilitating the interpretation of temporal correlation using R2 values.

After reading the authors response, and my own reading of the paper, I consider that the main comments were addressed. I'd like to thank the authors for providing a thorough response to reviewers, as such the paper can now be published.

Prof. Dr. Alexandre Langlois
Associate editor, *The Cryosphere*